# FAST PREDICTIVE UNCERTAINTY FOR CLASSIFICATION WITH BAYESIAN DEEP NETWORKS

## ABSTRACT

In Bayesian Deep Learning, distributions over the output of classification neural networks are approximated by first constructing a Gaussian distribution over the weights, then sampling from it to receive a distribution over the categorical output distribution. This is costly. We extend existing work to construct a Dirichlet approximation of this output distribution, yielding an analytic map between Gaussian distributions in logit space and Dirichlet distributions in the output space. We argue that the resulting Dirichlet distribution has theoretical and practical advantages, in particular more efficient computation of the uncertainty estimate, scaling to large datasets and networks like ImageNet and DenseNet. We demonstrate the use of this Dirichlet approximation by using it to construct a lightweight uncertainty-aware output ranking for the ImageNet setup.

## 1 INTRODUCTION

Quantifying the uncertainty of Neural Networks' (NNs) predictions is important in safety-critical applications such as medical-diagnosis (Begoli et al., 2019) and self-driving vehicles (McAllister et al., 2017; Michelmore et al., 2018). Architectures for classification tasks produce a probability distribution as their output, constructed by applying the softmax to the point-estimate output of the penultimate layer. However, it has been shown that this distribution is overconfident (Nguyen et al., 2015; Hein et al., 2019) and thus cannot be used for predictive uncertainty quantification.

Approximate Bayesian methods provide quantified uncertainty over the network's parameters and thus the outputs in a tractable fashion. The commonly used Gaussian approximate posterior (MacKay, 1992a; Graves, 2011; Blundell et al., 2015; Ritter et al., 2018) approximately induces a Gaussian distribution over the logits of a NN (Mackay, 1995). But the associated predictive distribution does not have an analytic form. It is thus generally approximated by Monte Carlo (MC) integration requiring multiple samples. Predictions in Bayesian Neural Networks (BNNs) are thus generally expensive operations.

In this paper, we re-consider an old but largely overlooked idea originally proposed by David JC MacKay (1998) in a different setting (arguably the inverse of the Deep Learning setting) which transforms a Dirichlet distribution into a Gaussian. Dirichlet distributions are generally defined on the simplex. But when its variable is defined on the inverse softmax's domain, its shape effectively approximates a Gaussian. The inverse of this approximation, which will be called the *Laplace Bridge* here (Hennig et al., 2012), analytically maps a Gaussian distribution onto a Dirichlet distribution. Given a Gaussian distribution over the logits of a NN, one can thus efficiently obtain an approximate Dirichlet distribution over the softmax outputs (Figure 1).

Our contributions are: We re-visit MacKay's derivation with particular attention to a symmetry constraint that becomes necessary in our "inverted" use of the argument from the Gaussian to the Dirichlet family. We then validate the quality of this approximation both theoretically and empirically, and demonstrate significant speed-up over MC-integration. Finally, we show a use-case, leveraging the analytic properties of Dirichlets to improve the popular top-$k$ metric through uncertainties.

Section 2 provides the mathematical derivation. Section 3 discusses the Laplace Bridge in the context of NNs. We compare it to the recent approximations of the predictive distributions of NNs in Section 4. Experiments are presented in Section 5.

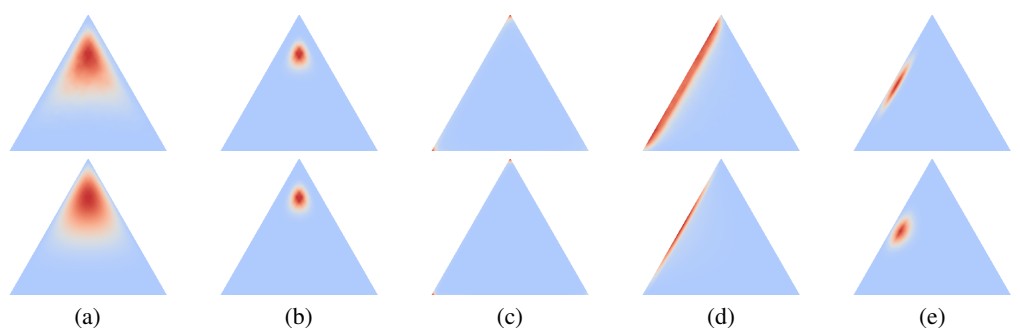

Figure 1: Densities on the simplex of the true distribution (top row, computed by MC integration) and "Laplace Bridge" approximation constructed in this paper (bottom row). For column (a) and (b), two different Gaussians were constructed, such that the resulting MAP estimate is the same, but the uncertainty differs. For (c), (d) and (e) the same mean with decreasing uncertainty was used. We find that in all cases the Laplace Bridge is a good approximation and captures the desired properties.

## 2 THE LAPLACE BRIDGE

Laplace approximations[1] are a popular and light-weight method to approximate a general probability distribution $q(\mathbf{x})$ with a Gaussian $\mathcal{N}(\mathbf{x}|\boldsymbol{\mu}, \boldsymbol{\Sigma})$ when $q(\mathbf{x})$ is twice differentiable and the Hessian at the mode is positive definite. It sets $\boldsymbol{\mu}$ to a mode of $q$, and $\boldsymbol{\Sigma} = -(\nabla^2 \log q(\mathbf{x})|_{\boldsymbol{\mu}})^{-1}$, the inverse Hessian of $\log q$ at that mode. This scheme can work well if the true distribution is unimodal and defined on the real vector space.

The Dirichlet distribution, which has the density function

$$\text{Dir}(\boldsymbol{\pi}|\boldsymbol{\alpha}) := \frac{\Gamma\left(\sum_{k=1}^{K} \alpha_k\right)}{\prod_{k=1}^{K} \Gamma(\alpha_k)} \prod_{k=1}^{K} \pi_k^{\alpha_k - 1} , \tag{1}$$

is defined on the probability simplex and can be "multimodal" in the sense that the distribution diverges in the $k$-corner of the simplex when $\alpha_k < 1$. Both issues preclude a Laplace approximation, at least in the naïve form described above. However, MacKay (1998) noted that both can be fixed, elegantly, by a change of variable (see Figure 2). Details of the following argument can be found in the supplements. Consider the $K$-dimensional variable $\boldsymbol{\pi} \sim \text{Dir}(\boldsymbol{\pi}|\boldsymbol{\alpha})$ defined as the softmax of $\mathbf{z} \in \mathbb{R}^K$:

$$\pi_k(\mathbf{z}) := \frac{\exp(z_k)}{\sum_{l=1}^{K} \exp(z_l)} , \tag{2}$$

for all $k = 1, \ldots, K$. We will call $\mathbf{z}$ the logit of $\boldsymbol{\pi}$. When expressed as a function of $\mathbf{z}$, the density of the Dirichlet in $\boldsymbol{\pi}$ has to be multiplied by the absolute value of the determinan of the Jacobian

$$\det \frac{\partial \boldsymbol{\pi}}{\partial \mathbf{z}} = \prod_k \pi_k(z), \tag{3}$$

thus removing the $-1$ terms in the exponent:

$$\text{Dir}_{\mathbf{z}}(\boldsymbol{\pi}(\mathbf{z})|\boldsymbol{\alpha}) := \frac{\Gamma\left(\sum_{k=1}^{K} \alpha_k\right)}{\prod_{k=1}^{K} \Gamma(\alpha_k)} \prod_{k=1}^{K} \pi_k(\mathbf{z})^{\alpha_k} , \tag{4}$$

This density of $\mathbf{z}$, the Dirichlet distribution in the *softmax basis*, can now be accurately approximated by a Gaussian through a Laplace approximation, yielding an analytic map from the parameter space $\boldsymbol{\alpha} \in \mathbb{R}_+^K$ to the parameter space of the Gaussian ($\boldsymbol{\mu} \in \mathbb{R}^K$ and symmetric positive definite $\boldsymbol{\Sigma} \in$

---

[1]For clarity: Laplace approximations are *also* one out of several possible ways to construct a Gaussian approximation to the weight posterior of a NN, by constructing a second-order Taylor approximation of the empirical risk at the trained weights. This is *not* the way they are used in this section. The Laplace Bridge is agnostic to how the input Gaussian distribution is constructed. It could, e.g., also be constructed as a variational approximation, or the moments of Monte Carlo samples. See also Section 5.

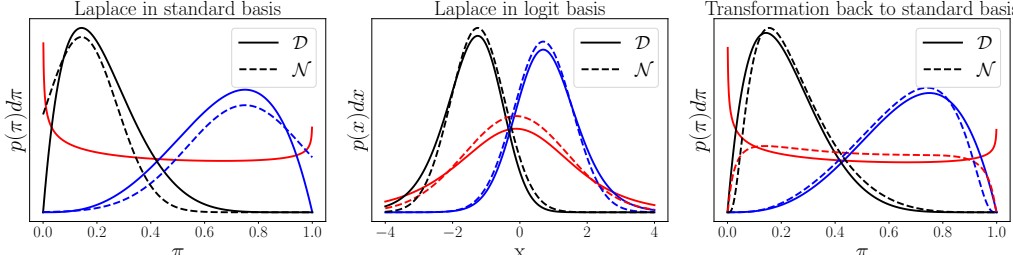

Figure 2: (Adapted from Hennig et al. (2012)). Visualization of the Laplace Bridge for the Beta distribution (1D special case of the Dirichlet). **Left:** "Generic" Laplace approximations of standard Beta distributions by Gaussians. Note that the Beta Distribution (red curve) does not even have a valid approximation because the Hessian is not positive semi-definite. **Middle:** Laplace approximation to the same distributions after basis transformation through the softmax (4). The transformation makes the distributions "more Gaussian" (i.e. uni-modal, bell-shaped, with support on the real line), thus making the Laplace approximation more accurate. **Right:** The same Beta distributions, with the back-transformation of the Laplace approximations from the middle figure to the simplex, yielding a much improved approximate distribution. In particular, in contrast to the left-most image, the dashed lines now actually are probability distributions (they integrate to 1 on the simplex).

$\mathbb{R}^{K \times K}$), given by

$$\mu_k = \log \alpha_k - \frac{1}{K} \sum_{l=1}^{K} \log \alpha_l \,, \tag{5}$$

$$\Sigma_{k\ell} = \delta_{k\ell} \frac{1}{\alpha_k} - \frac{1}{K} \left[ \frac{1}{\alpha_k} + \frac{1}{\alpha_\ell} - \frac{1}{K} \sum_{u=1}^{K} \frac{1}{\alpha_u} \right]. \tag{6}$$

The corresponding derivations require care because the Gaussian parameter space is evidently larger than that of the Dirichlet and not fully identified by the transformation. A pseudo-inverse of this map was provided by Hennig et al. (2012). It maps the Gaussian parameters to those of the Dirichlet as

$$\alpha_k = \frac{1}{\Sigma_{kk}} \left( 1 - \frac{2}{K} + \frac{e^{\mu_k}}{K^2} \sum_{l=1}^{K} e^{-\mu_l} \right) \tag{7}$$

(this equation ignores off-diagonal elements of $\Sigma$, more discussion below). Together, Eqs. 5, 6 and 7 will here be used for Bayesian Deep Learning, and jointly called the *Laplace Bridge*. A full derivation of the Laplace Bridge can be found in B and C. Even though the Laplace Bridge implies a reduction of the expressiveness of the distribution, we show in Section 3 that this map is still sufficiently accurate.

Figure 1 shows the quality of the resulting approximation. We consider multiple different $\mu, \Sigma$ in three dimensions. We exhaustively sample from the Gaussian and apply the softmax. The resulting histogram is compared to the PDF of the corresponding Dirichlet. The first part of the figure emphasizes that a point estimate is insufficient. Since the mean for the Dirichlet is the normalized parameter vector $\alpha$, the parameters ($\alpha_1 = [2, 2, 6]^\top$ and $\alpha_2 = [11, 11, 51]^\top$) yield the same point estimate even though their distributions are clearly different. The second part shows how the Laplace Bridge maps w.r.t decreasing uncertainty.

## 3 THE LAPLACE BRIDGE FOR BNNS

Let $f_\theta : \mathbb{R}^N \to \mathbb{R}^K$ be an $L$-layer Neural Network parametrized by $\theta \in \mathbb{R}^P$, with a Gaussian approximate posterior $\mathcal{N}(\theta|\mu_\theta, \Sigma_\theta)$. For any input $\mathbf{x} \in \mathbb{R}^N$, one way to obtain an approximate Gaussian distribution on the pre-softmax output (logit vector) $f_\theta(\mathbf{x}) =: \mathbf{z}$ is as

$$q(\mathbf{z}|\mathbf{x}) \approx \mathcal{N}(\mathbf{z}|f_{m\mu_\theta}(\mathbf{x}), \mathbf{J}(\mathbf{x})^\top \Sigma_\theta \mathbf{J}(\mathbf{x})) \,, \tag{8}$$

where $\mathbf{J}(\mathbf{x})$ is the $P \times K$ Jacobian matrix representing the derivative $\frac{\partial \mathbf{z}}{\partial \theta}$ (Mackay, 1995). This is a linearized approximation of the network which doesn't account for non-linearities and is often

computationally infeasible. For the experiments with larger networks we therefore use a last-layer Laplace approximation of the network as successfully used, e.g. by Snoek et al. (2015); Wilson et al. (2016); Brosse et al. (2020); Kristiadi et al. (2020). It is given by

$$q(\mathbf{z}|\mathbf{x}) \approx \mathcal{N}(\mathbf{z}|\mu_{W^{(l)}}\phi(\mathbf{x}), \phi(\mathbf{x})^T\Sigma_{W^{(l)}}\phi(\mathbf{x})), \tag{9}$$

where $\phi(x)$ denotes the output of a forward pass through the first $L-1$ layers, $\mu_{W^{(l)}}$ is the maximum likelihood estimate for the weights of the last layer, and $\Sigma_{W^{(l)}}$ is the inverse of the negative Hessian w.r.t. the loss $\Sigma_{W^{(l)}} = -\{\frac{\partial^2 \mathcal{L}}{\partial^2 W^{(l)}}\}^{-1}$.

Approximating the density of the softmax of this Gaussian random variable as a Dirichlet, using the Laplace Bridge, *analytically* approximates the predictive distribution in a single step, as opposed to many samples. From Eq. (7), this requires $\mathcal{O}(K)$ computations to construct the $K$ parameters $\alpha_k$ of the Dirichlet. In contrast, MC-integration has computational costs of $\mathcal{O}(MJ)$, where $M$ is the number of samples and $J$ is the cost of sampling from $q(\mathbf{z}|\mathbf{x})$ (typically $J$ is of order $K^2$ after an initial $\mathcal{O}(K^3)$ operation for a matrix decomposition of the covariance). The Monte Carlo approximation has the usual sampling error of $\mathcal{O}(1/\sqrt{M})$, while the Laplace Bridge has a fixed but small error (empirical comparison in Section 5.2).

We now discuss several qualitative properties of the Laplace Bridge relevant for the uncertainty quantification use case in Deep Learning. For output classes of "comparably high" probability, as defined below, the variance $\mathrm{Var}(\pi_k|\boldsymbol{\alpha})$ under the Laplace Bridge increases with the variance of the underlying Gaussian. In this sense, the Laplace Bridge approximates the uncertainty information encoded in the output of a BNN.

**Proposition 1** (proof in supplements). *Let* $\mathrm{Dir}(\boldsymbol{\pi}|\boldsymbol{\alpha})$ *be obtained via the Laplace Bridge from a Gaussian O* $\mathcal{N}(\mathbf{z}|\boldsymbol{\mu}, \boldsymbol{\Sigma})$ *over* $\mathbb{R}^K$. *Then, for each* $k = 1, \ldots, K$, *letting* $\alpha_{\neq k} := \sum_{l \neq k} \alpha_l$, *if*

$$\alpha_k > \frac{1}{4}\left(\sqrt{9\alpha_{\neq k}^2 + 10\alpha_{\neq k} + 1} - \alpha_{\neq k} - 1\right),$$

*then the variance* $\mathrm{Var}(\pi_k|\boldsymbol{\alpha})$ *of the $k$-th component of* $\boldsymbol{\pi}$ *is increasing in* $\boldsymbol{\Sigma}_{kk}$.

Intuitively, this result describes the condition that needs to be fulfilled such that the variance of the resulting Dirichlet scales with the variance of the $k$-th component of the Gaussian. It can be seen as a proxy for a high-quality approximation. An empirical evaluation showing that this condition is fulfilled in most cases can be found in the supplements.

Further benefits of this approximation arise from the convenient analytical properties of the Dirichlet exponential family. For example, a point estimate of the posterior predictive distribution is directly given by the Dirichlet's mean,

$$\mathbb{E}\boldsymbol{\pi} = \left(\frac{\alpha_1}{\sum_{l=1}^K \alpha_l}, \ldots, \frac{\alpha_K}{\sum_{l=1}^K \alpha_l}\right)^\top, \tag{10}$$

Additionally, Dirichlets have Dirichlet marginals: If $p(\boldsymbol{\pi}) = \mathrm{Dir}(\boldsymbol{\pi}|\boldsymbol{\alpha})$, then

$$p\left(\left[\pi_1, \pi_2, \ldots, \pi_j, \sum_{k>j}\pi_k\right]^\top\right) = \mathrm{Dir}\left(\alpha_1, \alpha_2, \ldots, \alpha_j, \sum_{k>j}\alpha_k\right). \tag{11}$$

An additional benefit of the Laplace Bridge for BNNs is that it is more flexible than an MC-integral. If we let $p(\boldsymbol{\pi})$ be the distribution over $\boldsymbol{\pi} := \mathrm{softmax}(\mathbf{z}) := [e^{z_1}/\sum_l e^{z_l}, \ldots, e^{z_K}/\sum_l e^{z_l}]^\top$, then the MC-integral can be seen as a "point-estimate" of this distribution since it approximates $\mathbb{E}\boldsymbol{\pi}$. In contrast, the Dirichlet distribution $\mathrm{Dir}(\boldsymbol{\pi}|\boldsymbol{\alpha})$ approximates the distribution $p(\boldsymbol{\pi})$. Thus, the Laplace Bridge enables tasks that can be done only with a distribution but not a point estimate. For instance, one could ask "what is the distribution of the first $L$ classes?" when one is dealing with $K$-class ($L < K$) classification. Since the marginal distribution can be computed analytically (11), the Laplace Bridge provides a convenient yet cheap way of answering this question.

## 4 RELATED WORK

In Bayesian Neural Networks, analytic approximations of posterior predictive distributions have attracted a great deal of research. In the binary classification case, for example, the probit approximation has been proposed already in the 1990s (Spiegelhalter and Lauritzen, 1990; MacKay,

1992b). However, while there exist some bounds (Titsias, 2016) and approximations of the expected log-sum-exponent function (Ahmed and Xing, 2007; Braun and McAuliffe, 2010), in the multi-class case, obtaining a good analytic approximation of the expected softmax function under a Gaussian measure is still considered an open problem. The Laplace Bridge can be used to produce a close analytical approximation of this integral. It furthers the trend of sampling-free solutions within Bayesian Deep Learning (Wu et al., 2018; Haussmann et al., 2019, etc.). The crucial difference is that, unlike these methods, the Laplace Bridge approximates the full distribution over the softmax outputs of a deep network.

Recently, it has been proposed to model the distribution of softmax outputs of a network directly. Similar to the Laplace Bridge, Malinin and Gales (2018; 2019); Sensoy et al. (2018) proposed to use the Dirichlet distribution to model the posterior predictive for non-Bayesian networks. They further proposed novel training techniques in order to directly learn the Dirichlet. In contrast, the Laplace Bridge tackles the problem of approximating the distribution over the softmax outputs of the ubiquitous Gaussian-approximated Bayesian networks (Graves, 2011; Blundell et al., 2015; Louizos and Welling, 2016; Sun et al., 2017, etc) without any additional training procedure. This allows the Laplace Bridge to be used with pre-trained networks and emphasizes its non-invasive nature.

## 5 Experiments

We conduct three main experiments. First, we compare the Laplace Bridge to the MC-integral in the example application of out-of-distribution (OOD) detection (Section 5.1). Secondly, we compare their computational cost and contextualize the speed-up for the prediction process in Section 5.2. Thirdly, in Section 5.3, we present analysis on ImageNet (Russakovsky et al., 2014) to demonstrate the scalability of the LB and the advantage of having a full Dirichlet distribution over softmax outputs. For a preliminary experiment to showcase useful features of the LB on the MNIST dataset and code for all experiments we refer the reader to Appendix D.

All experiments were conducted using different forms of Laplace approximations on the weight space. For the smaller experiments, a full (all-layer) Laplace approximation with a diagonal Hessian (LeCun et al., 1990) was used. For the experiments with larger networks, the Laplace approximation has been applied only to the last-layer of the network. This scheme has been successfully used by Snoek et al. (2015); Wilson et al. (2016); Brosse et al. (2020), etc. and it has been shown theoretically to mitigate overconfidence problems in ReLU networks (Kristiadi et al., 2020). For the last-layer experiments we use diagonal and Kronecker-factorized (Ritter et al., 2018) (KFAC) approximations of the Hessian, since inverting the exact Hessian is too costly. A detailed mathematical explanation and setup of the experiments can be found in appendix D. While the Laplace Bridge could also be applied to different approximations of a Gaussian posterior predictive such as Variational Inference (Graves, 2011; Blundell et al., 2015), we used a Laplace approximation in our experiments to construct such an approximation. This is for two reasons: (i) it is one of the fastest ways to get a Gaussian posterior predictive and (ii) it can be applied to pre-trained networks which is especially useful for ImageNet experiments. Nevertheless, we want to emphasize again that the Laplace Bridge can be applied to any Gaussian over the outputs independent of the way it was generated. Despite the overlap in nomenclature, the Laplace Bridge is *not* restricted to Gaussians arising from a Laplace approximation of the network.

### 5.1 OOD detection

We compare the performance of the Laplace Bridge to the MC-integral (Diagonal and KFAC) on a standard OOD detection benchmark suite, to test whether the Laplace Bridge gives similar results to the MC sampling methods and compare their computational overhead. Following prior literature, we use the standard mean-maximum-confidence (MMC) and area under the ROC-curve (AUROC) metrics (Hendrycks and Gimpel, 2016). For an in-distribution dataset, a higher MMC value is desirable while for the OOD dataset we want a lower MMC value (optimally, $1/K$ in $K$-class classification problems). For the AUROC metric, the higher the better, since it represents how good a method is for distinguishing in- and out-of-distribution datasets.

The test scenarios are: (i) A two-layer convolutional network trained on the MNIST dataset. To approximate the posterior over the parameter of this network, a full (all-layer) Laplace approximation with a diagonal Hessian is used. The OOD datasets for this case are FMNIST (Xiao et al.,

Table 1: OOD detection results. The Laplace Bridge (LB) wins most comparisons with Diagonal sampling and draws even with KFAC sampling w.r.t. both metrics. However, the LB is around 400 times faster on average which is the key property of the LB. 1000 samples were drawn from the Gaussian over the outputs motivated by Figure 3. The MNIST experiments were done with a Laplace approximation of the entire network while the others only used the last layer. The same table with error bars is in Appendix D.

| Train | Test | Diag Sampling | | Diag LB | | KFAC Sampling | | KFAC LB | | Time in s $\downarrow$ | |
|---|---|---|---|---|---|---|---|---|---|---|---|
| | | MMC $\downarrow$ | AUROC $\uparrow$ | MMC $\downarrow$ | AUROC $\uparrow$ | MMC $\downarrow$ | AUROC $\uparrow$ | MMC $\downarrow$ | AUROC $\uparrow$ | Sampling | LB |
| MNIST | MNIST | 0.942 | - | **0.987** | - | - | - | - | - | 26.8 | **0.062** |
| MNIST | FMNIST | 0.397 | 0.992 | **0.363** | **0.996** | - | - | - | - | 26.8 | **0.062** |
| MNIST | notMNIST | **0.543** | 0.960 | 0.649 | **0.961** | - | - | - | - | 50.3 | **0.117** |
| MNIST | KMNIST | **0.513** | **0.974** | 0.637 | 0.973 | - | - | - | - | 26.9 | **0.062** |
| CIFAR-10 | CIFAR-10 | 0.948 | - | **0.966** | - | 0.857 | - | **0.966** | - | 6.58 | **0.017** |
| CIFAR-10 | CIFAR-100 | **0.708** | **0.889** | 0.742 | 0.866 | 0.562 | 0.880 | 0.741 | 0.866 | 6.59 | **0.016** |
| CIFAR-10 | SVHN | **0.643** | 0.933 | 0.647 | **0.934** | 0.484 | 0.939 | 0.648 | 0.934 | 17.0 | **0.040** |
| SVHN | SVHN | 0.986 | - | **0.993** | - | 0.947 | - | **0.993** | - | 17.1 | **0.042** |
| SVHN | CIFAR-100 | 0.595 | 0.984 | **0.526** | 0.985 | 0.460 | 0.986 | 0.527 | 0.985 | 6.62 | **0.017** |
| SVHN | CIFAR-10 | 0.593 | 0.984 | **0.520** | 0.987 | 0.458 | 0.986 | 0.520 | 0.987 | 6.62 | **0.017** |
| CIFAR-100 | CIFAR-100 | **0.762** | - | 0.590 | - | **0.757** | - | 0.593 | - | 6.76 | **0.016** |
| CIFAR-100 | CIFAR-10 | 0.467 | 0.788 | **0.206** | **0.791** | 0.463 | 0.788 | 0.209 | 0.791 | 6.71 | **0.017** |
| CIFAR-100 | SVHN | 0.461 | 0.795 | **0.170** | **0.815** | 0.453 | 0.798 | 0.173 | 0.815 | 17.3 | **0.041** |

2017), notMNIST (Bulatov, 2011), and KMNIST (Clanuwat et al., 2018). (ii) For larger datasets, i.e. CIFAR-10 (Krizhevsky et al., 2014), SVHN (Netzer et al., 2011), and CIFAR-100 (Krizhevsky et al., 2014), we use a ResNet-18 network (He et al., 2016). Since this network is large, Equation (8) in conjunction with a full Laplace approximation is too costly. We, therefore, use a last-layer Laplace approximation to obtain the approximate diagonal and KFAC Gaussian posterior. The OOD datasets for CIFAR-10, SVHN, and CIFAR-100 are SVHN and CIFAR100; CIFAR-10 and CIFAR-100; and SVHN and CIFAR-10, respectively. In all scenarios, the networks are well-trained with 99% accuracy on MNIST, 95.4% on CIFAR-10, 76.6% on CIFAR-100, and 100% on SVHN. For the sampling baseline, we use 1000 posterior samples to compute the predictive distribution which is motivated by Figure 3. We use the mean of the Dirichlet to obtain a comparable approximation to the MC-integral. The results are presented in Table 1. The Laplace Bridge yields, on average, better results than diagonal sampling and ties with KFAC sampling w.r.t both metrics. However, the Laplace Bridge is around 400 times faster than both sampling-based methods and is therefore preferable.

Further comparisons with ensemble networks and other methods to approximate the integral of a softmax-Gaussian can be found in Appendix D. In both cases the LB is comparable or better in terms of OOD detection while being faster and yielding a full distribution.

## 5.2 TIME COMPARISON

We compare the computational cost of the density-estimated $p_{\text{sample}}$ distribution via sampling and the Dirichlet obtained from the Laplace Bridge $p_{\text{LB}}$ for approximating the true $p_{\text{true}}$ over softmax-Gaussian samples[2]. Different amounts of samples are drawn from the Gaussian, the softmax is applied and the KL divergence between the histogram of the samples with the true distribution is computed. We use KL-divergences $D_{\text{KL}}(p_{\text{true}}\|p_{\text{sample}})$ and $D_{\text{KL}}(p_{\text{true}}\|p_{\text{LB}})$, respectively, to measure similarity between approximations and ground truth while the number of samples for $p_{\text{sample}}$ is increased exponentially. The true distribution $p_{\text{true}}$ is constructed via MC with 100k samples. The experiment is conducted for three different Gaussian distributions over $\mathbb{R}^3$. Since the softmax applied to a Gaussian does not have an analytic form, the algebraic calculation of the approximation error is not possible and an empirical evaluation via sampling is the best option. The logistic-Normal (Aitchison and Shen, 1980) is a different distribution and not useful for our purposes as it has no analytic expected value. The fact that there is no analytic solution is part of the justification for using the Laplace Bridge in the first place.

---

[2]I.e. samples are obtained by first sampling from a Gaussian and transforming it via the softmax function.

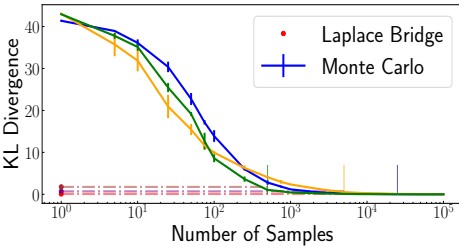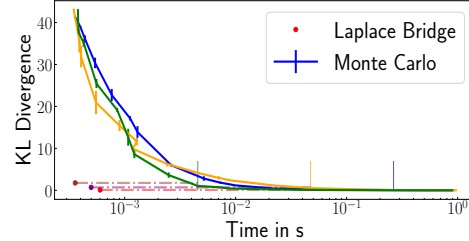

Figure 3: KL-divergence plotted against the number of samples (left) and wall-clock time (right). Monte Carlo density estimation becomes as good as the Laplace Bridge after around 750 to 10000 samples and takes at least 100 times longer. The three lines represent three different samples.

Figure 3 suggests that the number of samples required such that the distribution $p_{\text{sample}}$ approximates the true distribution $p_{\text{true}}$ as good as the Dirichlet distribution obtained via the Laplace Bridge is large, i.e. somewhere between 500 and 10000. This translates to a wall-clock time advantage of at least a factor of 100 before sampling becomes competitive in quality with the Laplace Bridge.

To better understand how light-weight and non-invasive the Laplace Bridge is, we have timed different parts of the process for our setup. Training a ResNet-18 on CIFAR10 over 130 epochs takes us 71 minutes and 30 seconds. Computing a Hessian for the network from the training data can be done with (Dangel et al., 2019) at the cost of one backward pass or around 29 seconds. No other changes to the training procedure are necessary to apply the LB. Since the Laplace Bridge only applies to

Table 2: Contextualization of the timings for the entire predictive process. We find that the LB provides a significant speed-up of the process as a whole.

| # samples in brackets | Forward pass | MC(1000) | MC(100) | MC(10) | Laplace Bridge |
|---|---|---|---|---|---|
| Time in seconds | $0.448 \pm 0.004$ | $8.183 \pm 0.080$ | $0.867 \pm 0.004$ | $0.117 \pm 0.001$ | $0.014 \pm 0.000$ |
| Fraction of overall time | 0.05/0.34/0.79/0.97 | 0.95 | 0.66 | 0.21 | 0.03 |

the last step of the prediction pipeline it is important to compare its time to that of a forward pass through the rest of the network. Re-using the ResNet-18 and CIFAR10 setup we measure the time in seconds for a forward pass, for the application of the Laplace Bridge, and for the sampling procedure with 10, 100, and 1000 samples. The resulting sum of timings for the entire test set is given in Table 2). We find that sampling is between 95% (for 1000 samples) and 21% (for 10 samples) of the entire prediction while the LB is only 3%. These results show that the time accelerations through the LB are a meaningful improvement for the prediction process as a whole.

## 5.3 Uncertainty-aware output ranking on ImageNet

Classification tasks on large datasets with many classes, like ImageNet, are not often done in a Bayesian fashion since the posterior inference and sampling are expensive. The Laplace Bridge, in conjunction with the last-layer Bayesian approximations, can be used to alleviate this problem. Furthermore, having a full distribution over the softmax outputs of a BNN gives rise to new possibilities. For example, one could subsume all classes which have sufficiently overlapping marginal distributions into one if they are semantically similar as illustrated in Figure 4.

Another possibility is to improve the standard classification metrics. Large classification tasks like ImageNet are often compared along a top-5 metric, i.e. it is tested whether the correct class is within the five most probable estimates of the network. Although widely accepted, this metric has some pathologies. Consider two examples: i) Assume the network has to classify a hypothetical image of "raptor" and it is confident that the label is either a "hawk" or an "eagle". Then all probability mass should be distributed between those two classes. ii) Assume the network has to classify an image of which it is confident that it is a "fish" but it is uncertain between ten different possible fish species. In both cases a static rule (always 5) is ill-suited.

Leveraging the probabilistic output provided by the Laplace Bridge, we propose a simple decision rule that can handle such examples and is more fine-grained due to its awareness of uncertainty. One may call such a rule *uncertainty-aware top-$k$*; pseudocode for the algorithm is given in the

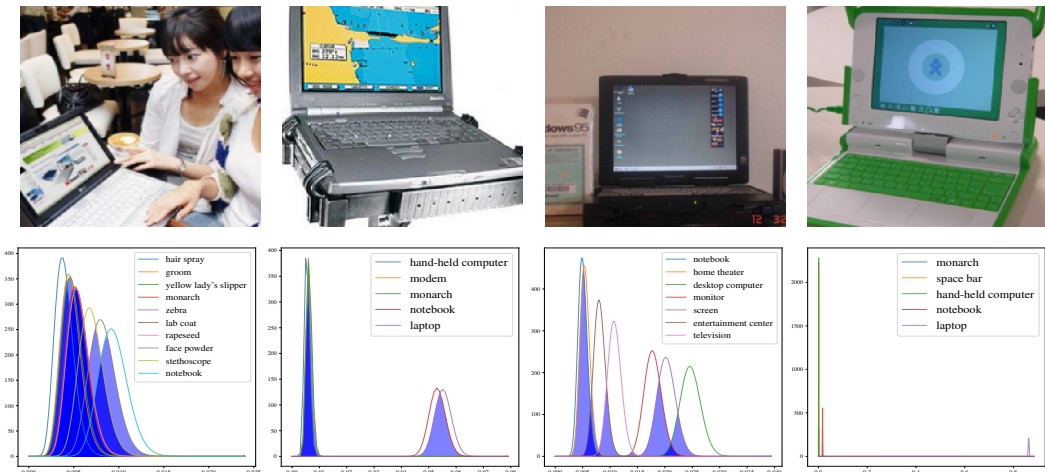

Figure 4: **Upper row:** images from the "laptop" class of ImageNet. **Bottom row:** Beta marginals of the top-$k$ predictions for the respective image. In the first column, the overlap between the marginal of all classes is large, signifying high uncertainty, i.e. the prediction is "I do not know". In the second column, "notebook" and "laptop" have confident, yet overlapping marginal densities and therefore yield a top-2 prediction: "either a notebook or a laptop". In the third column "desktop computer", "screen" and "monitor" have overlapping marginal densities, yielding a top-3 estimate. The last case shows a top-1 estimate: the network is confident that "laptop" is the only correct label.

supplements. Instead of taking the top-$k$ as a decision threshold for an arbitrary $k$ we take the uncertainty/confidence of the model to inform the decision. This is more flexible and therefore able to handle situations in which different numbers of classes are plausible outcomes. The Dirichlet distribution obtained from the Laplace Bridge provides this capability. In particular, since the marginal distribution over each component of a Dirichlet distribution is a $\mathrm{Beta}(\alpha_i, \sum_{j\neq i} \alpha_j)$, this can be done analytically and efficiently. The proposed decision rule uses the area of overlap between the marginal distributions of the sorted outcomes. This is similar to hypotheses testing, i.e. $t$-tests (Nickerson, 2000) or its Bayesian alternatives (Masson, 2011). If, for example, two Beta densities overlap more than $5\%$, we cannot say that they are different distributions with high confidence. All distributions that have sufficient overlap should become the new top-$k$ estimate. Pseudo code for the uncertainty aware top-k ranking can be found in Algorithm 1. Figure 4 shows four examples from the "laptop" class of ImageNet. We evaluate this decision rule on the test set of ImageNet. The

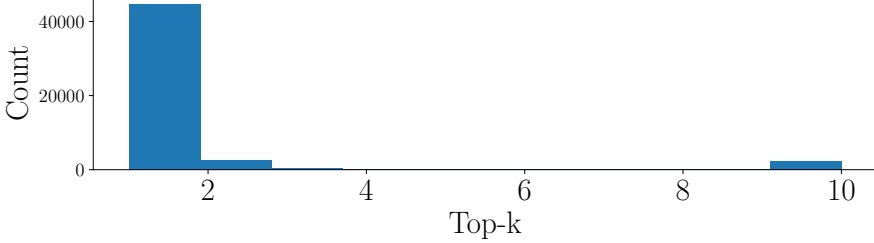

Figure 5: A histogram of ImageNet predictions' length using the proposed uncertainty-aware top-$k$. Most test images are a top-1 prediction, indicating high confidence. There are some top-2, top-3, and top-10 predictions, showing increased uncertainty. All predictions $> 10$ are grouped in the 10-bin for visibility.

overlap is calculated through the inverse CDF[3] of the respective Beta marginals. The original top-1 accuracy of DenseNet on ImageNet is $0.744$. Meanwhile, the uncertainty-aware top-$k$ accuracy is $0.797$, where $k$ is on average $1.688$. A more detailed analysis of the distribution of top-$k$ estimates (see Figure 5) shows that most of the predictions given by the uncertainty-aware metric still yielded a top-1 prediction. This means that using uncertainty does not imply adding meaningless classes to the prediction. However, there are some non-negligible cases where $k$ equals to 2, 3, or 10 where all

---

[3]Also known as the quantile function or percent point function

values larger than 10 are in the 10 bin for visibility. This indicates that whenever the class labels are ambiguous, our method can detect it, and thus yields a significantly higher accuracy.

---

**Algorithm 1** Uncertainty-aware top-$k$

---

**Input:** A Dirichlet parameter $\boldsymbol{\alpha} \in \mathbb{R}^K$ obtained by applying the Laplace Bridge to the Gaussian over the logit of an input, a percentile threshold $T$ e.g. $0.05$, a function $\mathrm{class\_of}$ that returns the underlying class of a sorted index.

$\tilde{\boldsymbol{\alpha}} = \mathrm{sort\_descending}(\boldsymbol{\alpha})$          *// start with the highest confidence*
$\alpha_0 = \sum_i \alpha_i$
$\mathcal{C} = \{\mathrm{class\_of}(1)\}$          *// initialize top-k, must include at least one class*
**for** $i = 2, \ldots, K$ **do**
    $F_{i-1} = \mathrm{Beta}(\tilde{\alpha}_{i-1}, \alpha_0 - \tilde{\alpha}_{i-1})$          *// the previous marginal CDF*
    $F_i = \mathrm{Beta}(\tilde{\alpha}_i, \alpha_0 - \tilde{\alpha}_i)$          *// the current marginal CDF*
    $l_{i-1} = F_{i-1}^{-1}(T/2)$          *// left $\frac{T}{2}$ percentile of the previous marginal*
    $r_i = F_i^{-1}(1 - T/2)$          *// right $\frac{T}{2}$ percentile of the current marginal*
    **if** $r_i > l_{i-1}$ **then**
        $\mathcal{C} = \mathcal{C} \cup \{\mathrm{class\_of}(i)\}$          *// overlap detected, add the current class*
    **else**
        **break**          *// No more overlap, end the algorithm*
    **end if**
**end for**

**Output:** $\mathcal{C}$          *// return the resulting top-k prediction*

---

## 6  CONCLUSION

We have adapted an old but overlooked approximation scheme for new use in Bayesian Deep Learning. Given a Gaussian approximation to the weight-space posterior of a neural network (which can be constructed by various means, including another Laplace approximation), and an input, the Laplace Bridge analytically maps the marginal Gaussian prediction on the logits onto a Dirichlet distribution over the softmax vectors. The associated computational cost of $\mathcal{O}(K)$ for $K$-class prediction compares favorably to that of Monte Carlo sampling. The proposed method both theoretically and empirically preserves predictive uncertainty, offering an attractive, low-cost, high-quality alternative to Monte Carlo sampling. In conjunction with a low-cost, last-layer Bayesian approximation, it can be useful in real-time applications wherever uncertainty is required—especially because it reduces the cost of predicting a posterior distribution at test time.

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

## A    APPENDIX

### FIGURES

The parameters of Figure 1 are the $\mu$ and $\Sigma$ mapped from $\alpha = (2, 2, 6)$ and $\alpha = (11, 11, 51)$ through the Laplace Bridge for a) and b) respectively. The values have been chosen to have the same mode of the Dirichlet distribution but clearly yield different uncertainties. For c), d) and e) they share $\mu = (-1, 2, -1)^\top$ and vary $\Sigma$ with $10 \cdot I_3$, $1 \cdot I_3$ and $0.1 \cdot I_3$ where $I_3$ is the three-dimensional identity matrix.

The parameters of Figure 2 are from left to right $\alpha, \beta = (0.8, 0.9), (4, 2, ), (2, 7)$.

### CHANGE OF VARIABLE FOR PDF

Let $\mathbf{z}$ be an $n$-dimensional continuous random variable with joint density function $p_{\mathbf{x}}$. If $\mathbf{y} = G(\mathbf{x})$, where $G$ is a differentiable function, then $\mathbf{y}$ has density $p_{\mathbf{y}}$:

$$p(\mathbf{y}) = f\left(G^{-1}(\mathbf{y})\right) \left| \det \left[ \frac{dG^{-1}(\mathbf{z})}{d\mathbf{z}} \bigg|_{\mathbf{z}=\mathbf{y}} \right] \right| \tag{12}$$

where the differential is the Jacobian of the inverse of $G$ evaluated at $\mathbf{y}$. This procedure, also known as 'change of basis', is at the core of the Laplace bridge since it is used to transform the Dirichlet into the softmax basis.

### PROOF FOR PROPOSITION

*Proof.* Considering that $\alpha_k$ is a decreasing function of $\Sigma_{kk}$ by definition (7), it is sufficient to show that under the hypothesis, the derivative of $\frac{\partial}{\partial \alpha_k} \mathrm{Var}(\pi_k | \boldsymbol{\alpha})$ is negative.

By definition, the variance $\mathrm{Var}(\pi_k | \boldsymbol{\alpha})$ is

$$\mathrm{Var}(\pi_k | \boldsymbol{\alpha}) = \frac{\frac{\alpha_k}{\alpha_k + \alpha_{\neq k}} - \frac{\alpha_k^2}{(\alpha_k + \alpha_{\neq k})^2}}{\alpha_k + \alpha_{\neq k} + 1} .$$

The derivative is therefore

$$\frac{\partial}{\partial \alpha_k} \mathrm{Var}(\pi_k | \boldsymbol{\alpha}) =$$

$$\frac{\alpha_{\neq k}(\alpha_{\neq k}^2 - \alpha_{\neq k}\alpha_k + \alpha_{\neq k} - \alpha_k(2\alpha_k + 1))}{(\alpha_k + \alpha_{\neq k})^3 (\alpha_k + \alpha_{\neq k} + 1)^2} .$$

Solving $\frac{\partial}{\partial \alpha_k} \mathrm{Var}(\pi_k | \boldsymbol{\alpha}) < 0$ for $\alpha_k$ yields

$$\alpha_k > \frac{1}{4} \left( \sqrt{9\alpha_{\neq k}^2 + 10\alpha_{\neq k} + 1} - \alpha_{\neq k} - 1 \right) .$$

Therefore, under this hypothesis, $\mathrm{Var}(\pi_k | \boldsymbol{\alpha})$ is a decreasing function of $\alpha_k$. $\qquad \square$

### EXPERIMENTAL EVALUATION OF THE PROPOSITION

To test how often the condition is fulfilled we count its frequency. The fact that the condition is fulfilled implies a good approximation. The fact that the condition is not fulfilled does not automatically imply a bad approximation.

## B    APPENDIX (DERIVATION OF LB)

Assume we have a Dirichlet in the standard basis with parameter vector $\boldsymbol{\alpha}$ and probability density function:

$$\mathrm{Dir}(\boldsymbol{\pi}|\boldsymbol{\alpha}) := \frac{\Gamma\left(\sum_{k=1}^K \alpha_k\right)}{\prod_{k=1}^K \Gamma(\alpha_k)} \prod_{k=1}^K \pi_k^{\alpha_k - 1} , \tag{13}$$

We aim to transform the basis of this distribution via the softmax transform to be in the new base $\pi$:

| | | ‖ | frequency |
|---|---|---|---|
| MNIST | MNIST | ‖ | - |
| MNIST | FMNIST | ‖ | - |
| MNIST | notMNIST | ‖ | - |
| MNIST | KMNIST | ‖ | - |
| CIFAR-10 | CIFAR-10 | ‖ | 0.998 |
| CIFAR-10 | CIFAR-100 | ‖ | 0.925 |
| CIFAR-10 | SVHN | ‖ | 0.832 |
| SVHN | SVHN | ‖ | 0.999 |
| SVHN | CIFAR-100 | ‖ | 0.668 |
| SVHN | CIFAR-10 | ‖ | 0.653 |
| CIFAR-100 | CIFAR-100 | ‖ | 0.662 |
| CIFAR-100 | CIFAR-10 | ‖ | 0.214 |
| CIFAR-100 | SVHN | ‖ | 0.166 |

Table 3

$$\pi_k(\mathbf{z}) := \frac{\exp(z_k)}{\sum_{l=1}^{K} \exp(z_l)}, \tag{14}$$

Usually, to transform the basis we would need the inverse transformation $H^{-1}(\mathbf{z})$ as described in the main paper. However, the softmax does not have an analytic inverse. Therefore David JC MacKay uses the following trick. Assume we know that the distribution in the transformed basis is:

$$\mathrm{Dir}_{\mathbf{z}}(\boldsymbol{\pi}(\mathbf{z})|\boldsymbol{\alpha}) := \frac{\Gamma\left(\sum_{k=1}^{K} \alpha_k\right)}{\prod_{k=1}^{K} \Gamma(\alpha_k)} \prod_{k=1}^{K} \pi_k(\mathbf{z})^{\alpha_k}, \tag{15}$$

then we can show that the original distribution is the result of the basis transform by the softmax.

**The Dirichlet in the softmax basis:** We show that the density over $\boldsymbol{\pi}$ shown in Equation 15 transforms into the Dirichlet over $\mathbf{z}$. First, we consider the special case where $\boldsymbol{\pi}$ is confined to a $I - 1$ dimensional subspace satisfying $\sum_i \pi_i = c$. In this subspace we can represent $\boldsymbol{\pi}$ by an $I - 1$ dimensional vector $\mathbf{a}$ such that

$$\pi_i = \mathbf{a}_i \quad i, ..., I - 1 \tag{16}$$

$$\pi_I = c - \sum_i^{I-1} \mathbf{a}_i \tag{17}$$

and similarly we can represent $\mathbf{z}$ by an $I - 1$ dimensional vector $\varphi$:

$$z_i = \varphi_i \quad i, ..., I - 1 \tag{18}$$

$$z_I = 1 - \sum_i^{I-1} \varphi_i \tag{19}$$

then we can find the density over $\varphi$ (which is proportional to the required density over $\mathbf{z}$) from the density over $\boldsymbol{\pi}$ (which is proportional to the given density over $\boldsymbol{\pi}$) by finding the determinant of the $(I - 1) \times (I - 1)$ Jacobian $\mathbf{J}$ given by

$$J_{ik} = \frac{\partial z_i}{\partial \varphi_i} = \sum_j^{I} \frac{\partial z_i}{\partial \pi_j} \frac{\partial \pi_j}{\partial \varphi_k}$$

$$= \delta_{ik}\mathbf{z}_i - \mathbf{z}_i\mathbf{z}_k + \mathbf{z}_i\mathbf{z}_I = \mathbf{z}_i(\delta_{ik} - (\mathbf{z}_k - \mathbf{z}_I)) \tag{20}$$

We define two additional $I-1$ dimensional helper vectors $\mathbf{z}_k^+ := \mathbf{z}_k - \mathbf{z}_I$ and $n_k := 1$, and use $\det(I - xy^T) = 1 - x \cdot y$ from linear algebra. It follows that

$$
\begin{aligned}
\det J &= \prod_{i=1}^{I-1} \mathbf{z}_i \times \det[I - n\mathbf{z}^{+^T}] \\
&= \prod_{i=1}^{I-1} \mathbf{z}_i \times (1 - n \cdot \mathbf{z}^+) \\
&= \prod_{i=1}^{I-1} \mathbf{z}_i \times \left(1 - \sum_k \mathbf{z}_k^+\right) = I \prod_{i=1}^{I} \mathbf{x}_i
\end{aligned}
\tag{21}
$$

Therefore, using Equation 15 we find that

$$
P(\mathbf{z}) = \frac{P(\boldsymbol{\pi})}{|\det \mathbf{J}|} \propto \prod_{i=1}^{I} \mathbf{z}_i^{\alpha_i - 1}
\tag{22}
$$

This result is true for any constant $c$ since it can be put into the normalizing constant. Thereby we make sure that the integral of the distribution is 1 and we have a valid probability distribution.

## C  APPENDIX (DERIVATION OF INVERSION)

Through the figures of the 1D Dirichlet approximation in the main paper we have already established that the mode of the Dirichlet lies at the mean of the Gaussian distribution and therefore $\boldsymbol{\pi}(\mathbf{y}) = \frac{\alpha}{\sum_i \alpha_i}$. Additionally, the elements of $\mathbf{y}$ must sum to zero. These two constraints combined yield only one possible solution for $\boldsymbol{\mu}$.

$$
\mu_k = \log \alpha_k - \frac{1}{K} \sum_{l=1}^{K} \log \alpha_l
\tag{23}
$$

Calculating the covariance matrix $\boldsymbol{\Sigma}$ is more complicated but layed out in the following. The logarithm of the Dirichlet is, up to additive constants

$$
\log p_y(y|\alpha) = \sum_k \alpha_k \pi_k
\tag{24}
$$

Using $\pi_k$ as the softmax of $\mathbf{y}$ as shown in Equation 14 we can find the elements of the Hessian $\mathbf{L}$

$$
L_{kl} = \hat{\alpha}(\delta_{kl} \hat{\pi}_k - \hat{\pi}_k \hat{\pi}_l)
\tag{25}
$$

where $\hat{\boldsymbol{\alpha}} := \sum_k \alpha_k$ and $\hat{\boldsymbol{\pi}} = \frac{\alpha_k}{\hat{\alpha}}$ for the value of $\boldsymbol{\pi}$ at the mode. Analytically inverting $\mathbf{L}$ is done via a lengthy derivation using the fact that we can write $\mathbf{L} = \mathbf{A} + \mathbf{X}\mathbf{B}\mathbf{X}^\top$ and inverting it with the Schur-complement. This process results in the inverse of the Hessian

$$
L_{kl}^{-1} = \delta_{kl} \frac{1}{\alpha_k} - \frac{1}{K} \left[ \frac{1}{\alpha_k} + \frac{1}{\alpha_l} - \frac{1}{K} \left( \sum_{u}^{K} \frac{1}{\alpha_u} \right) \right]
\tag{26}
$$

We are mostly interested in the diagonal elements, since we desire a sparse encoding for computational reasons and we otherwise needed to map a $K \times K$ covariance matrix to a $K \times 1$ Dirichlet parameter vector which would be a very overdetermined mapping. Note that $K$ is a scalar not a matrix. The diagonal elements of $\boldsymbol{\Sigma} = \mathbf{L}^{-1}$ can be calculated as

$$
\Sigma_{kk} = \frac{1}{\alpha_k} \left(1 - \frac{2}{K}\right) + \frac{1}{K^2} \sum_{l}^{k} \frac{1}{\alpha_l}.
\tag{27}
$$

To invert this mapping we transform Equation 23 to

$$
\alpha_k = e^{\mu_k} \prod_{l}^{K} \alpha_l^{1/K}
\tag{28}
$$

by applying the logarithm and re-ordering some parts. Inserting this into Equation 27 and re-arranging yields

$$\prod_l^K \alpha_l^{1/K} = \frac{1}{\mathbf{\Sigma}_{kk}} \left[ e^{-\mu} \left( 1 - \frac{2}{K} \right) + \frac{1}{K^2} \sum_u^K e^{-\mu_u} \right] \tag{29}$$

which can be re-inserted into Equation 28 to give

$$\alpha_k = \frac{1}{\Sigma_k k} \left( 1 - \frac{2}{K} + \frac{e^{-\mu_k}}{K^2} \sum_l^K e^{-\mu_k} \right) \tag{30}$$

which is the final mapping. With Equations 23 and 27 we are able to map from Dirichlet to Gaussian and with Equation 30 we are able to map the inverse direction.

## D  APPENDIX (EXPERIMENTAL DETAILS)

The exact experimental setups, i.e. network architectures, learning rates, random seeds, etc. can be found in the accompanying anonymized GitHub repository [4]. This section is used to justify some of the decisions we made during the process in more detail, highlight some miscellaneous interesting things and showcase the additional experiments promised in the main paper.

### EXPERIMENTAL SETUP

The (F-, K-, not-)MNIST networks have been trained with Adam, a learning rate of $1e-6$ and batch size of 32 over 6 epochs. The batch size was chosen to be so low because the Jacobian for the full-layer Laplace approximation would otherwise not fit into memory. CIFAR10, CIFAR100 and SVHN have been trained with SGD, a start learning rate of $0.1$, momentum of $0.9$, weight decay of $5e-4$ and a batch size of 128. We applied a learning rate schedule that multiplies the learning rate with $0.1$ at epoch 30, 80 and 130 for CIFAR10, at 100, 150, 200 for CIFAR100 and at 10, 20 and 25 for SVHN. (F-, K-, not-)MNIST has been trained on a small 2 conv-layer + 1 linear layer setup and all other OOD experiments were conducted with a ResNet-18. For the ImageNet experiments we used the pre-trained weights from PyTorch and only computed the Hessian information with BACKpack (Dangel et al., 2019) in one backward pass. All prior uncertainties for the Laplace approximation have been chosen such that the resulting MMC is not more than $5-10\%$ away from the MAP estimate to have comparable results.

### MATHEMATICAL DESCRIPTION OF THE SETUP

In principle, the Gaussian over the weights required by the Laplace Bridge for BNNs can be constructed by any Gaussian approximate Bayesian method such as variational Bayes (Graves, 2011; Blundell et al., 2015) and Laplace approximations for NNs (MacKay, 1992a; Ritter et al., 2018). We will focus on the Laplace approximation, which uses the same principle as the Laplace Bridge. However, in the Laplace approximation for neural networks, the posterior distribution over the weights of a network is the one that is approximated as a Gaussian, instead of a Dirichlet distribution over the outputs as in the Laplace Bridge.

Given a dataset $\mathcal{D} := \{(\mathbf{x}_i, t_i)\}_{i=1}^D$ and a prior $p(\boldsymbol{\theta})$, let

$$p(\boldsymbol{\theta}|\mathcal{D}) \propto p(\boldsymbol{\theta})p(\mathcal{D}|\boldsymbol{\theta}) = p(\boldsymbol{\theta}) \prod_{(\mathbf{x},t)\in\mathcal{D}} p(y=t|\boldsymbol{\theta},\mathbf{x}), \tag{31}$$

be the posterior over the parameter $\boldsymbol{\theta}$ of an $L$-layer network $f_{\boldsymbol{\theta}}$. Then we can get an approximation of the posterior $p(\boldsymbol{\theta}|\mathcal{D})$ by fitting a Gaussian $\mathcal{N}(\boldsymbol{\theta}|\boldsymbol{\mu_\theta}, \boldsymbol{\Sigma_\theta})$ where

$$\boldsymbol{\mu_\theta} = \boldsymbol{\theta}_{\text{MAP}},$$
$$\boldsymbol{\Sigma_\theta} = (-\nabla^2|_{\boldsymbol{\theta}_{\text{MAP}}} \log p(\boldsymbol{\theta}|\mathcal{D}))^{-1} =: \mathbf{H}_{\boldsymbol{\theta}}^{-1}.$$

That is, we fit a Gaussian centered at the mode $\boldsymbol{\theta}_{\text{MAP}}$ of $p(\boldsymbol{\theta}|\mathcal{D})$ with the covariance determined by the curvature at that point. We assume that the prior $p(\boldsymbol{\theta})$ is a zero-mean isotropic Gaussian $\mathcal{N}(\boldsymbol{\theta}|\mathbf{0}, \sigma^2\mathbf{I})$ and the likelihood function is the Categorical density

$$p(\mathcal{D}|\boldsymbol{\theta}) = \prod_{(\mathbf{x},t)\in\mathcal{D}} \text{Cat}(y=t|\text{softmax}(f_{\boldsymbol{\theta}}(\mathbf{x}))).$$

---

[4] https://github.com/19219181113/LB_for_BNNs

For various applications in Deep Learning, an approximation with full Hessian is often computationally too expensive. Indeed, for each input $\mathbf{x} \in \mathbb{R}^N$, one has to do $K$ backward passes to compute the Jacobian $\mathbf{J}(\mathbf{x})$. Moreover, it requires an $\mathcal{O}(PK)$ storage which is also expensive since $P$ is often in the order of millions. A cheaper alternative is to fix all but the last layer of $f_{\boldsymbol{\theta}}$ and only apply the Laplace approximation on $\mathbf{W}_L$, the last layer's weight matrix. This scheme has been used successfully by Snoek et al. (2015); Wilson et al. (2016); Brosse et al. (2020), etc. and has been shown theoretically that it can mitigate overconfidence problems in ReLU networks (Kristiadi et al., 2020). In this case, given the approximate last-layer posterior

$$p(\mathbf{W}^L|\mathcal{D}) \approx \mathcal{N}(\text{vec}(\mathbf{W}^L)|\text{vec}(\mathbf{W}_{\text{MAP}}^L), \mathbf{H}_{\mathbf{W}^L}^{-1}), \tag{32}$$

one can efficiently compute the distribution over the logits. That is, let $\boldsymbol{\phi} : \mathbb{R}^N \to \mathbb{R}^Q$ be the first $L-1$ layers of $f_{\boldsymbol{\theta}}$, seen as a feature map. Then, for each $\mathbf{x} \in \mathbb{R}^N$, the induced distribution over the logit $\mathbf{W}^L \boldsymbol{\phi}(\mathbf{x}) =: \mathbf{z}$ is given by

$$p(\mathbf{z}|\mathbf{x}) = \mathcal{N}(\mathbf{z}|\mathbf{W}_{\text{MAP}}^L \boldsymbol{\phi}(\mathbf{x}), (\boldsymbol{\phi}(\mathbf{x})^\top \otimes \mathbf{I}) \mathbf{H}_{\mathbf{W}^L}^{-1} (\boldsymbol{\phi}(\mathbf{x}) \otimes \mathbf{I})), \tag{33}$$

where $\otimes$ denotes the Kronecker product.

An even more efficient last-layer approximation can be obtained using a Kronecker-factored matrix normal distribution (Louizos and Welling, 2016; Sun et al., 2017; Ritter et al., 2018). That is, we assume the posterior distribution to be

$$p(\mathbf{W}^L|\mathcal{D}) \approx \mathcal{MN}(\mathbf{W}^L|\mathbf{W}_{\text{MAP}}^L, \mathbf{U}, \mathbf{V}), \tag{34}$$

where $\mathbf{U} \in \mathbb{R}^{K \times K}$ and $\mathbf{V} \in \mathbb{R}^{Q \times Q}$ are the Kronecker factorization of the inverse Hessian matrix $\mathbf{H}_{\mathbf{W}^L}^{-1}$ (Martens and Grosse, 2015). In this case, for any $\mathbf{x} \in \mathbb{R}^N$, one can easily show that the distribution over logits is given by

$$p(\mathbf{z}|\mathbf{x}) = \mathcal{N}(\mathbf{z}|\mathbf{W}_{\text{MAP}}^L \boldsymbol{\phi}(\mathbf{x}), (\boldsymbol{\phi}(\mathbf{x})^\top \mathbf{V} \boldsymbol{\phi}(\mathbf{x})) \mathbf{U}), \tag{35}$$

which is easy to implement and computationally cheap. Finally, and even more efficient, is a last-layer approximation scheme with a diagonal Gaussian approximate posterior, i.e. the so-called mean-field approximation. In this case, we assume the posterior distribution to be

$$p(\mathbf{W}^L|\mathcal{D}) \approx \mathcal{N}(\text{vec}(\mathbf{W}^L)|\text{vec}(\mathbf{W}_{\text{MAP}}^L), \text{diag}(\boldsymbol{\sigma}^2)), \tag{36}$$

where $\boldsymbol{\sigma}^2$ is obtained via the diagonal of the Hessian of the log-posterior w.r.t. $\text{vec}(\mathbf{W}^L)$ at $\text{vec}(\mathbf{W}_{\text{MAP}}^L)$.

UNCERTAINTY ESTIMATES ON MNIST

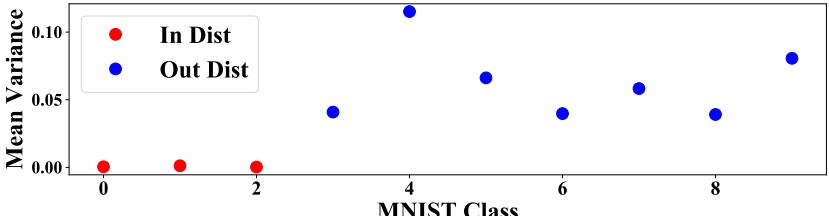

Figure 6: Average variance of the Dirichlet distributions of each MNIST class. The in-distribution uncertainty (variance) is nearly nil, while out-of-distribution variance is higher. This implies usability for OOD detection.

To give a good preliminary intuitive understanding of the LBs usefulness we empirically investigate the approximation quality of the Laplace Bridge in a "real-world" BNN on the MNIST dataset. A CNN with 2 convolutional and 2 fully-connected layers is trained on the first three digits of MNIST (the digits 0, 1, and 2). Adam optimizer with learning rate 1e-3 and weight decay 5e-4 is used. The batch size is 128. To obtain the posterior over the weights of this network, we perform a full (all-layer) Laplace approximation using Back-PACK (Dangel et al., 2019) to get the diagonal Hessian. The network is then evaluated on the full test set of MNIST (containing all ten classes). We present the results in Figure 6. We show for each $k = 1, \dots, K$, the average variance $\frac{1}{D_k} \sum_{i=1}^{D_k} \text{Var}(\pi_k(f_{\boldsymbol{\theta}}(\mathbf{x}_i)))$ of the resulting Dirichlet distribution over the softmax outputs, where $D_k$ is the number of test points predicted with label $k$. The results show that the variance of the Dirichlet distribution obtained via the Laplace Bridge is useful for uncertainty quantification: OOD data can be easily detected since the mean variance of the first three classes is nearly zero while that of the others is higher.

OOD DETECTION

Every experiment has been conducted with 5 different seeds. In the tables the mean and standard deviations are presented. The reason why the sampling procedure for the CIFAR-10 and CIFAR-100 case are similarly

fast even though we draw from a 10- vs 100-dimensional Gaussian is because the sampling procedures were parallelized on a GPU. All prior uncertainties over the weights were chosen such that the MMC of the sampling averages was around 5% lower than the MAP estimate.

To compare the OOD detection capabilities of the LB with other more standard methods we first evaluate it compared to an ensemble of 100 different sets of weights. As shown in Table 4 we find that the Laplace Bridge yields comparable results while being more than a thousand times faster.

Table 4: Results for sampling from all weights instead of just the last layer. Number of samples was 100. Sampling all weights and doing a forward pass seems to be worse than using the Laplace Bridge even though it takes much longer.

| Train | Test | Sampling (100) | | | Laplace Bridge | | |
|---|---|---|---|---|---|---|---|
| | | MMC ↓ | AUROC ↑ | Time in s ↓ | MMC ↓ | AUROC ↑ | Time in s ↓ |
| MNIST | MNIST | $0.989 \pm 0.000$ | - | 76.3 | $\mathbf{0.987} \pm 0.000$ | - | **0.062** |
| MNIST | FMNIST | $0.465 \pm 0.004$ | $0.992 \pm 0.000$ | 76.3 | $\mathbf{0.363} \pm 0.000$ | $\mathbf{0.996} \pm 0.000$ | **0.062** |
| MNIST | notMNIST | $\mathbf{0.645} \pm 0.001$ | $0.955 \pm 0.001$ | 42.6 | $0.649 \pm 0.000$ | $\mathbf{0.961} \pm 0.000$ | **0.117** |
| MNIST | KMNIST | $\mathbf{0.632} \pm 0.001$ | $0.967 \pm 0.001$ | 76.0 | $0.637 \pm 0.000$ | $\mathbf{0.973} \pm 0.000$ | **0.062** |

However, we think the Laplace Bridge should primarily not be compared to other OOD detection algorithms such as those listed in Hendrycks and Gimpel (2016) but rather to algorithms that approximate the integral of the softmax transform applied to Gaussian samples since they have comparable speed. The two algorithms we compare to the LB are the Extended MacKay approach Gibbs (1997) (see equation 5.33 in (Gibbs)) which uses

$$\int \frac{\exp(a^{(j)})}{\sum_i \exp(a^{(i)})} \frac{1}{Z} \exp\left[-\frac{1}{2}\sum_i \frac{(a^{(i)} - \bar{a}^{(i)})^2}{v^{(i)}}\right] \approx \frac{\exp(\tau(v^{(j)})\bar{a}^{(j)})}{\sum_i \exp(\tau(v^{(i)})\bar{a}^{(i)})} \tag{37}$$

as an approximation. The logit output of class $i$ is denoted by $a^{(i)}$, its average by $\bar{a}^{(i)}$, its variance by $v^{(i)}$ and $\tau(v) = 1/\sqrt{1 + \pi \cdot v/8}$. The second approximation is from Wu et al. (2018) (equation 23 in appendix) and we will refer to it as the "second-order delta posterior predictive (SODPP)". It is given by

$$p(y) \approx p \odot \left[1 + p^\top \Sigma p - \Sigma p + \frac{1}{2}\text{diag}(\Sigma) - \frac{1}{2}\text{diag}(\Sigma)\right] \tag{38}$$

where $p$ is the vector of logit outputs, $\Sigma$ is its covariance matrix, and $\odot$ represents the entry-wise product. The results in Table 5 suggest that the LB shows slightly better performance than the other two approaches. Additionally, the LB not only yields an approximation of the integral but additionally a fully parameterized Dirichlet distribution of the output classes and is therefore preferable.

Table 5: Comparison of the Laplace Bridge Dirichlet mean with other methods to compute the integral over a softmax-Gaussian. We find that the LB is slightly better than the other two methods while additionally not only providing an estimate for the integral but also giving a fully parameterized Distribution.

| Train | Test | Extended MacKay | | SODPP | | Laplace Bridge | |
|---|---|---|---|---|---|---|---|
| | | MMC ↓ | AUROC ↑ | MMC ↓ | AUROC ↑ | MMC ↓ | AUROC ↑ |
| MNIST | MNIST | $0.981 \pm 0.001$ | - | $0.979 \pm 0.001$ | - | $\mathbf{0.987} \pm 0.001$ | - |
| MNIST | FMNIST | $0.557 \pm 0.022$ | $0.985 \pm 0.002$ | $0.546 \pm 0.024$ | $0.983 \pm 0.003$ | $\mathbf{0.491} \pm 0.035$ | $\mathbf{0.990} \pm 0.002$ |
| MNIST | notMNIST | $0.737 \pm 0.015$ | $0.928 \pm 0.013$ | $\mathbf{0.708} \pm 0.017$ | $0.929 \pm 0.013$ | $0.754 \pm 0.012$ | $\mathbf{0.933} \pm 0.012$ |
| MNIST | KMNIST | $0.673 \pm 0.006$ | $\mathbf{0.967} \pm 0.002$ | $\mathbf{0.642} \pm 0.006$ | $0.965 \pm 0.003$ | $0.710 \pm 0.008$ | $\mathbf{0.967} \pm 0.003$ |
| CIFAR-10 | CIFAR-10 | $\mathbf{0.970} \pm 0.001$ | - | $\mathbf{0.970} \pm 0.001$ | - | $0.963 \pm 0.001$ | - |
| CIFAR-10 | CIFAR-100 | $0.801 \pm 0.003$ | $\mathbf{0.873} \pm 0.002$ | $0.793 \pm 0.010$ | $0.872 \pm 0.002$ | $\mathbf{0.756} \pm 0.003$ | $0.860 \pm 0.001$ |
| CIFAR-10 | SVHN | $0.714 \pm 0.036$ | $\mathbf{0.933} \pm 0.012$ | $0.710 \pm 0.036$ | $0.932 \pm 0.012$ | $\mathbf{0.638} \pm 0.049$ | $0.931 \pm 0.017$ |
| SVHN | SVHN | $0.976 \pm 0.002$ | - | $0.985 \pm 0.002$ | - | $\mathbf{0.993} \pm 0.000$ | - |
| SVHN | CIFAR-100 | $0.566 \pm 0.007$ | $\mathbf{0.984} \pm 0.002$ | $\mathbf{0.508} \pm 0.008$ | $\mathbf{0.984} \pm 0.002$ | $0.616 \pm 0.015$ | $0.981 \pm 0.004$ |
| SVHN | CIFAR-10 | $0.563 \pm 0.007$ | $\mathbf{0.985} \pm 0.002$ | $\mathbf{0.505} \pm 0.009$ | $\mathbf{0.985} \pm 0.002$ | $0.618 \pm 0.017$ | $0.981 \pm 0.004$ |
| CIFAR-100 | CIFAR-100 | $0.748 \pm 0.000$ | - | $\mathbf{0.758} \pm 0.000$ | - | $0.593 \pm 0.000$ | - |
| CIFAR-100 | CIFAR-10 | $0.440 \pm 0.000$ | $0.790 \pm 0.000$ | $0.455 \pm 0.000$ | $0.787 \pm 0.000$ | $\mathbf{0.209} \pm 0.000$ | $\mathbf{0.791} \pm 0.000$ |
| CIFAR-100 | SVHN | $0.427 \pm 0.000$ | $0.802 \pm 0.000$ | $0.450 \pm 0.000$ | $0.793 \pm 0.000$ | $\mathbf{0.173} \pm 0.000$ | $\mathbf{0.815} \pm 0.000$ |

TIME COMPARISON

Every experiment has been conducted with 5 different seeds. The presented curves are the averages over these 5 experiments with errorbars. The reason why taking one sample is slower than two is because of the way random numbers are generated for the normal distribution. For further information read up on the Box-Mueller Transform.

Table 6: OOD detection results. Same table as in the main experiments but with error estimates. The Laplace Bridge (LB) wins most comparisons with Diagonal sampling and draws even with KFAC sampling w.r.t. both metrics. However, the LB is around 400 times faster on average. 1000 samples were drawn from the Gaussian over the outputs motivated by Figure 3. The (F-, K-, not-)MNIST experiments were done with a Laplace approximation of the entire network while the others only used the last layer. Five runs with different seeds per experiment were conducted. Standard deviation is reported to the third decimal.

| | | Diag Sampling | | Diag LB | | KFAC Sampling | | KFAC LB | | Time in s $\downarrow$ | |
|---|---|---|---|---|---|---|---|---|---|---|---|
| Train | Test | MMC $\downarrow$ | AUROC $\uparrow$ | MMC $\downarrow$ | AUROC $\uparrow$ | MMC $\downarrow$ | AUROC $\uparrow$ | MMC $\downarrow$ | AUROC $\uparrow$ | Sampling | LB |
| MNIST | MNIST | $0.942 \pm 0.007$ | - | **$0.987 \pm 0.000$** | - | - | - | - | - | 26.8 | **0.062** |
| MNIST | FMNIST | $0.397 \pm 0.001$ | $0.992 \pm 0.000$ | **$0.363 \pm 0.000$** | **$0.996 \pm 0.000$** | - | - | - | - | 26.8 | **0.062** |
| MNIST | notMNIST | **$0.543 \pm 0.000$** | $0.960 \pm 0.000$ | $0.649 \pm 0.000$ | **$0.961 \pm 0.000$** | - | - | - | - | 50.3 | **0.117** |
| MNIST | KMNIST | **$0.513 \pm 0.001$** | **$0.974 \pm 0.000$** | $0.637 \pm 0.000$ | $0.973 \pm 0.000$ | - | - | - | - | 26.9 | **0.062** |
| CIFAR-10 | CIFAR-10 | $0.948 \pm 0.000$ | - | **$0.966 \pm 0.000$** | - | $0.857 \pm 0.003$ | - | **$0.966 \pm 0.000$** | - | 6.58 | **0.017** |
| CIFAR-10 | CIFAR-100 | **$0.708 \pm 0.000$** | **$0.889 \pm 0.000$** | $0.742 \pm 0.000$ | $0.866 \pm 0.000$ | **$0.562 \pm 0.003$** | **$0.880 \pm 0.012$** | $0.741 \pm 0.000$ | $0.866 \pm 0.000$ | 6.59 | **0.016** |
| CIFAR-10 | SVHN | **$0.643 \pm 0.000$** | $0.933 \pm 0.000$ | $0.647 \pm 0.000$ | **$0.934 \pm 0.000$** | **$0.484 \pm 0.004$** | **$0.939 \pm 0.001$** | $0.648 \pm 0.003$ | $0.934 \pm 0.001$ | 17.0 | **0.040** |
| SVHN | SVHN | $0.986 \pm 0.000$ | - | **$0.993 \pm 0.000$** | - | $0.947 \pm 0.002$ | - | **$0.993 \pm 0.000$** | - | 17.1 | **0.042** |
| SVHN | CIFAR-100 | $0.595 \pm 0.000$ | $0.984 \pm 0.000$ | **$0.526 \pm 0.000$** | **$0.985 \pm 0.000$** | **$0.460 \pm 0.004$** | **$0.986 \pm 0.001$** | $0.527 \pm 0.002$ | $0.985 \pm 0.000$ | 6.62 | **0.017** |
| SVHN | CIFAR-10 | $0.593 \pm 0.000$ | $0.984 \pm 0.000$ | **$0.520 \pm 0.000$** | **$0.987 \pm 0.000$** | **$0.458 \pm 0.004$** | $0.986 \pm 0.001$ | $0.520 \pm 0.002$ | **$0.987 \pm 0.000$** | 6.62 | **0.017** |
| CIFAR-100 | CIFAR-100 | **$0.762 \pm 0.000$** | - | $0.590 \pm 0.000$ | - | **$0.757 \pm 0.000$** | - | $0.593 \pm 0.000$ | - | 6.76 | **0.016** |
| CIFAR-100 | CIFAR-10 | $0.467 \pm 0.000$ | $0.788 \pm 0.000$ | **$0.206 \pm 0.000$** | **$0.791 \pm 0.000$** | $0.463 \pm 0.000$ | $0.788 \pm 0.000$ | **$0.209 \pm 0.000$** | **$0.791 \pm 0.000$** | 6.71 | **0.017** |
| CIFAR-100 | SVHN | $0.461 \pm 0.000$ | $0.795 \pm 0.000$ | **$0.170 \pm 0.000$** | **$0.815 \pm 0.000$** | $0.453 \pm 0.001$ | $0.798 \pm 0.001$ | **$0.173 \pm 0.000$** | **$0.815 \pm 0.000$** | 17.3 | **0.041** |

UNCERTAINTY-AWARE OUTPUT RANKING ON IMAGENET

The prior covariances for the Laplace approximation of the Hessian over the weights were chosen such that uncertainty estimate of the Laplace bridge MMC over the outputs was not more than 5% lower than the MAP estimate. The length of list generated by our uncertainty aware method was chosen such that it contained at least one and maximally ten samples. Originally we wanted to choose the maximal length according to the size of the largest category (e.g. fishes or dogs) but the class tree hierarchy of ImageNet does not answer this question meaningfully. We chose ten because there are no reasonable bins larger than ten when looking at a histogram. Pseudo code for the uncertainty aware top-k ranking can be found in Algorithm 1.

