# OpenReview forum: "Fast Predictive Uncertainty for Classification with Bayesian Deep Networks"
_ICLR.cc/2021/Conference — Reject_

### Official Review · AnonReviewer2 · 2020-10-28

**Rating:** 4
**Confidence:** 5

**Review:**

This paper propose to use Laplace bridge as a building block to map the distribution of logits to the distribution of post-softmax predictions. The authors also provide some theoretical analysis on when such Laplace approximation behaves reasonably.

The idea of constructing a Dirichlet distribution to describe the output distribution for classification in Bayesian neural networks is not new. For example, natural-parameter networks (NPN) [1] support exponential family distributions including Dirichlet distributions as the output of each layer. The idea of Dirichlet distributions is extended and made more explicit in [2,3].

The idea of using the Laplace bridge in BNN is interesting and can potentially be useful. I like the demonstration in Figure 1, showing that Laplace bridge is a good approximation.

One of my major concerns is that a large trunk of related works are missing, especially those directly related to the use of Dirichlet distributions in BNN’s output. Lack of comparison and description makes it difficult and potentially misleading for reviewers to evaluate the proposed method’s technical merit. Besides, another line of highly related works is on combining Laplace approximation and BNN, such as [5].

Eq. 8 is incorrect. Since f_theta is an L-layer network, with the parameter theta, the mean of the network’s output is not simply \mu_theta^T x. This is because of the multi-layer structure with nonlinearity. Therefore Eq. 8 only holds when L=1.

Another potentially misrepresentation of related works is that: note that David MacKay’s proposed BNN treatment also involves Laplace approximation. Given that the Laplace bridge idea also originated from David MacKay, it would be better to clarify such related works and the differences between the proposed method and David MacKay’s works.

Another point that is unclear from the paper is the computation of the Hessian matrix in a large neural network. Given that a neural network typically has a large number of parameters, it would be important to provide details on related implementation in practice.

The authors emphasize one of the advantages of the proposed method is that it is sampling-free and therefore significantly speeds up computation. Note that there is a rich literature of sampling-free BNN (e.g., [1,2,3,4]) in recent years.

Although results on Table 2 make sense, the table itself is a bit misleading. (1) Even with LB, one still need a FF pass to compute the prediction. (2) the 29-second overhead should be made clear in the caption.

The experiment in Figure 4 is interesting, but it is a bit unclear what the point is and whether similar conclusion can be drawn using non-Bayesian NN, since softmax also provides uncertainty estimates for classification tasks naturally. Alternatively, it would be helpful to demonstrate the proposed method’s advantages in terms of obtaining calibrated uncertainty using metrics such as ECE or Brier Score.

As I said, this could potentially be a very interesting paper if related works are handled and compared to thoroughly, especially highly relevant ideas proposed before [1,2,3]. Otherwise it is difficult to evaluate the actual novelty and advance this paper provides.

Minor:
Typos in the caption of Figure 4.

[1] Natural-Parameter Networks: A Class of Probabilistic Neural Networks, NIPS 2016
[2] Feed-forward Propagation in Probabilistic Neural Networks with Categorical and Max Layers, ICLR 2018
[3] Sampling-free Epistemic Uncertainty Estimation Using Approximated Variance Propagation, CVPR 2019
[4] Probabilistic Backpropagation for Scalable Learning of Bayesian Neural Networks, ICML 2015
[5] Being Bayesian, Even Just a Bit, Fixes Overconfidence in ReLU Networks, ICML 2020

---

> ### Author Response · Authors · 2020-11-11
> **Adressing comments of R2**
>
> Thank you for your review. Since the majority of your criticism was directed at the comparison with related work we want to address it first.
>
> - **Related work - Dirichlet and BNNs:** We agree that using Dirichlets to model the output of BNNs is not new as we have acknowledged in the paper. Thanks for pointing out related work. Note that we already cite Kristiadi et al. [5] in the paper. We are uncertain how relevant the other suggestions are to our work. Maybe you could clarify. Paper [1]  is another example of the use of Dirichlets in BNNs and we can add it as such but is otherwise unrelated. Paper [2] uses categoricals of which the Dirichlet is a conjugate prior but the method seems to be otherwise unrelated. Paper [3] is a further example of sampling-free literature and we can cite it as such but is otherwise unrelated to our paper. Paper [4] is a different training technique that is also in the realm of BNNs but seems otherwise unrelated to our paper. Maybe you could clarify which kind of discussion of [1-4] you would expect to improve the paper in your eyes.
> In comparison to your suggestions, we think that there are still some major advantages of the LB for BNNs. The LB for BNNs does not require a new architecture or retraining of the weights. You can use your standard, well-working training kit and apply the LB without any loss of accuracy. As far as we understand it, all of [1-4] change either the architecture or the training procedure which, in both cases, means new hyperparameter search, new problems during training, etc. These are problems that the LB for BNNs does not share. We will add this to the discussion of the related work after your replies.
> - **Equation 8:** There is indeed a typo in equation 8 and it should be $\mathcal{N}(f(x, \theta)| f_{\mu_\theta}(x), J^T \Sigma J)$. This is a Laplace approximation of a linearized network. We have additionally added a formal statement about the Gaussian from the last-layer Laplace approximation that we use in the more complex experiments which is given by $\mathcal{N}(\mu \phi(x), \phi(x)^T \Sigma \phi(x))$ where $\phi(x)$ denotes a forward pass of $x$ through all $N-1$ layers previous to the last. Does this clarify your concerns with equation 8?
> - **Distinction to MacKay:** While MacKay provided the forward direction to the Laplace Bridge, the inverse direction was not developed until 2012 and has not been used in the context of Neural Networks to our knowledge. Additionally, we are the first to provide theoretical (see Proposition 1) and practical (see Figure 3 & Table 1) error estimates for the application of the LB to BNNs. Since they are necessary to convince other users to apply the technique we think this is sufficiently important and novel to be a meaningful distinction to MacKay’s work. For more see the response about novelty to R3.
> - **Computation of the Hessian:** We mention at several points in the paper that we use diagonal and KFAC approximation of the Hessian to reduce computational costs. However, maybe this is not what you meant with the comment. Could you please clarify? Additionally, we use last-layer Laplace approximations of the network for large networks because they have shown to yield similar performances as full-layer Laplace approximations at a fraction of the cost (Snoek et al. (2015); Wilson et al. (2016); Brosse et al. (2020), etc. ).
> - **Table 2 & speed-up:** The numbers in the table tell the entire story and the 29 seconds overhead are not part of the forward pass during prediction. The 29-second computation of the Hessian has to be done once during training. For the prediction/application of the network, the timings in table 2 are the timings for the entire process! This really is the selling point of this method because it provides a full output distribution essentially for free during test time.
> - **Figure 4 & top-k:** Theoretically, similar estimates could also be done without a Bayesian framework. However, as work cited in the introduction (Nguyen et al.,2015; Hein et al., 2019) shows, non-Bayesian uncertainties are often overconfident and therefore yield badly calibrated estimates. Using the Bayesian framework fixes this (see [5]).
>
> Thank you for your cooperative style. If you think we have misunderstood parts of your review we would be grateful for clarification. We hope our response has addressed all your concerns about our work.

---

### Official Review · AnonReviewer4 · 2020-10-28
**Good and simple idea, but more work needs to be done**

**Rating:** 6
**Confidence:** 5

**Review:**

The authors propose an approach called the Laplace Bridge to approximate predictive uncertainty in Bayesian neural networks. The approach is essentially based on first a change of variable, followed by a Laplace approximation. They provided a theoretical result for this approach, which essentially shows that for \alpha_k large enough, the variance of \pi_k given \alpha is increasing with the variance \Sigma_kk.  They performed some experiments to essentially show the computational speedup of Laplace Bridge against more cumbersome MC based competitors.

The pros of this approach is obvious: it is remarkably simple, and provides a sizable speed up computationally when compared to the cumbersome existing tools for Bayesian Neural Networks, and it uses well-established tools such as the Dirichlet and Laplace approximation.  The authors did a good job explaining the motivation and the paper is written in a clear manner. The experiments appear to demonstrate that this should work well in practice.

There remains, however, several cons that I believe the authors need to address:

First, the author should comment on how this is a *substantial* and *sufficiently novel* contribution to the field? The Laplace Bridge idea/derivations seem to come, for the most part, from Mckay and Hennig et al’s  (2012) etc prior work. The Laplace approximation idea is a common tool, and using the Dirichlet distribution for uncertainty quantification in neural networks for classification has been amply explored in other prior work (e.g. Sensoy 2018, Malinin etc). To play Devil’s advocate, it does seem like the authors are just applying to the BNN context something that we know works fast (Laplace Approximation) and comparing it to something that we know works slow (MC Sampling).

Second, the authors should try to justify with more rigor/guarantees on how good the approximation is. I do think the computational speed up in the empirical evaluations are big. However, currently, it appears that the Laplace Bridge is simply a heuristic, with minimal quantification/guarantees on the approximation error etc. For example, in Figure 2 (Right), in the very simple case of approximating a beta distribution (transformed back to the standard basis) , one sees that near 0 and 1, the approximation appears to be doing very poorly. So currently, the justification that Laplace Bridge is a good approximation comes mainly from visual plots (figure 2) and empirical evaluations. More rigor in the quantification of approximation error would be ideal.

Minor comments:
In the first line of section 2, “...light-weight method to approximate a general probability distribution…”, I believe that q has to satisfy certain smoothness conditions, so the phrase “a general probability distribution” appears to be too general.

Overall, the idea is simple and appears to lead to substantial speedups in practice/experiments, and the paper is written clearly , but authors should address issues of 1. where the substantial novelty/contribution comes from and 2. more rigorous/quantitive guarantees for approximation error.

---

> ### Author Response · Authors · 2020-11-11
> **Adressing Novelty and Approximation error with R4**
>
> Thank you for your clear and helpful review. As your structure is very clear we will dive right into our responses.
> - **Minor comment:** yes, q(y) has to be twice differentiable and the Hessian at the mode has to be positive definite. We have added this sentence.
> - **Major 1 - Novelty:**  While MacKay provided the forward direction to the Laplace Bridge, the inverse direction was not developed until 2012 and has not been used in the context of Neural Networks to our knowledge.
> We provided a long answer to the novelty question in our discussion with AnonReviewer 3 (see above: Concern 1 - Novelty). They also address all of your concerns about novelty.
>  - **Major 2 - Approximation error:** We share your desire for theoretical error analysis of the Laplace Bridge in the context of BNNs. However, as the softmax-Gaussian (which is not equal to the logistic-Gaussian; see R1) has no analytic pdf or integral, we can’t derive an analytic computation of the error. We are therefore forced to compute the error empirically (as in Figure 3) and investigate it visually (as in Figures 1 & 2) instead of giving hard bounds. It is also arguably more useful than a theoretical bound because it provides a direct comparison (in runtime and quality) to the competing method, sampling, thus quantifying not just *that* the method is good, but also showing *how* good it is compared to the alternative. For somebody who considers using the LB, we thought that this might be more convincing than a very abstract statement about the tightness of error bounds.
>
> Thank you for your cooperative style. If you think we have misunderstood parts of your review we would be grateful for clarification. We hope our response has addressed all your concerns about our work.

---

### Official Review · AnonReviewer3 · 2020-10-29

**Rating:** 5
**Confidence:** 4

**Review:**

Summary:
This article improves the efficiency of Bayesian neural networks (BNNs). It follows the sampling-free solutions within Bayesian deep learning (Wu et al., 2018; Haussmann et al., 2019, etc.). The difference is the proposed Laplace Bridge that approximates the  full distribution over the softmax outputs of a neural network. The Laplace Bridge tackles the problem of approximating the distribution over the softmax outputs of the ubiquitous Gaussian-approximated BNNs without any additional training  procedure. This allows the Laplace Bridge to be used with pre-trained networks and emphasizes its non-invasive nature.

Major comments:
- Overall, I find the paper is easy to follow and the experimental evaluation shows promising results. But my major concern is about the novelty of this work, given the fact that the basic idea was originally proposed by MacKay (1998) in a different  setting which transforms a Dirichlet distribution into a Gaussian and the inverse of this approximation, called the Laplace Bridge, was already proposed in the literature (Hennig et al., 2012).
- I would recommend the authors to provide more details on the training of BNNs within the Laplace Bridge, including the hyperparameters used for the training stage.

Minor comments:
- It is recommended to further check with the wording, e.g., the line 4 of the title of Figure 4 “In the column” and adding the description of units of x/y axis.
- The size of some figures appears too small, for example Fig. 4, which may hinder readability.

At the moment, I recommend a weak reject as the main weakness is the novelty, but I could be open to increasing my score if my concerns are addressed.

---

> ### Author Response · Authors · 2020-11-11
> **Adressing Novelty and Hyperparameters with R3**
>
> Thank you for your feedback. Both of your criticisms are pretty clear so we will just answer them in chronological order.
> - **Concern 1 - Novelty:** While we agree that our paper is to some extent based on previous work, there are several novel aspects. Just to outline a few: a) The inverse direction of the LB as described in our paper has so far not been used in the Neural Network context. b) We provide new analysis of the error estimate that shows, both in theory (see Proposition 1), and in practice (see Figure 3 & Table 1) that the LB provides high-quality estimates at a fraction of the cost of alternative estimates. c) With the top-k ranking we provide a new use-case for Dirichlet output distributions that provides meaningful uncertainty quantification for ImageNet at a cost that no other BNN method for Dirichlets can match because we can reuse pre-trained weights. Note that nearly all other BNN approaches stay away from Imagenet because it is not computationally feasible.
> Most importantly though, 22 years after the MacKay paper and 8 years after the Hennig paper, to our knowledge, nobody within the ML community seems to use the Laplace Bridge for their BNNs. Most people apply costly sampling even though they could use the Laplace Bridge to save time and get a parameterized Dirichlet instead of just samples. It is thus important to bring this “old” solution to renewed attention. We argue that we do so while also adding the new content stated above.
> - **Concern 2 - Hyperparameters:** There is a link to an anonymized GitHub repository in the appendix of the paper where you can find the code to all experiments that we ran including the seeds, learning rates, batch sizes, etc. We have additionally added a section called **Experimental Setup** in Appendix D which includes all training details. If you think we should include further information we will include them.
> - **Minor Comments:** We have updated the caption of figure 4 and removed the typo. Unfortunately, the margins of the ICLR template are so large that this was the best possible size and placement of figure 4 that we could find after many tries. If you have an additional suggestion we will try it.
>
> Thank you for your cooperative style. If you think we have misunderstood parts of your review we would be grateful for clarification. We hope our response has addressed all your concerns about our work.

---

### Official Review · AnonReviewer1 · 2020-10-30
**A nice revival of softmax basis ideas from MacKay (1998) and Henning et al (2012), could be above bar if glaring presentation issues fixed**

**Rating:** 5
**Confidence:** 4

**Review:**

Review Summary
--------------

Overall this paper offers a simple idea -- the laplacian bridge, revived from earlier work by MacKay (1998) and Henning et al (2012) -- that produces useful yet affordable predictive posterior estimates. I thought the fundamental approach is sound and does seem to be an elegant way to apply some older ideas that do indeed seem to be overlooked. My chief concerns are that the presentation contains many distracting errors (e.g. as written Eq 8, the primary equation of their method, is fundamentally wrong but I think this is a typo) and many steps lack justification. I think the core idea is promising, and a strong author response might convince me that the paper should be accepted.

Paper Summary
-------------

The paper presents the idea of the "Laplacian Bridge", a deterministic mapping provided in Eqs. 5-7 for converting between the parameters of a K-dimensional multivariate Normal distribution and the parameters of a K-dimensional Dirichlet distribution in the *softmax* basis. The mapping is "pseudo"-invertible (e.g. it works from Normal to Dirichlet and from Dirichlet to Normal, but only approximately). The Dirichlet to Normal direction is derived by performing a Laplace approximation to a Dirichlet written in the softmax basis. As cited, changing the parameterization to softmax is an ideal from MacKay in the 1990s. The alternative direction (Normal to Dirichlet) in Eq. 7 is a construction due to Henning et al (2012), where given a known mean parameter \mu (a K-dim vector) and covariance \Sigma (a KxK matrix), we can obtain a Dirichlet concentration parameter vector \alpha (a vector of size K).

While this abstract construction is known, the present paper uses it specifically to approximate the posterior predictive of a multi-class Bayesian neural network. Given an approximate posterior q(w|\mu, \Sigma) over weights, we

The approximate posterior of pre-softmax activations z (real valued) given input features x is:

    p(z | x) = \int_w p(z | x, w) q(w | \mu, \Sigma) dw
    		 \approx N( z | m, S)

with expressions for mean m and covariance S given in Equation 8. Note that even after determining parameters m and S, in order to make predictions at input x we need to draw several samples of vector z, feed each through a softmax, and average the results. Drawing a single sample of vector z from a normal with covariance S usually has cost O(K^2).

This paper advocates for the following approximation of the post-softmax activations \pi(z):

	p( \pi(z) | x) = Dirichlet( laplace_bridge(m, S) )

where the parameters of the dirichlet are determined by the laplace bridge construction in Eq. 7.

The contribution is that the expected value of the class probabilities \pi under the Dirichlet has closed form, so predictions can be made without *any* sampling, just by computing the posterior mean. The total prediction cost is O(K) once m and S are determined.

One new theoretical result is presented in proposition 1, showing that under some conditions, the variance of the k-th entry of a probability vector constructed using the LB method increases with \Sigma_kk (the relevant variance of this entry under the Normal).

The first set of results (Sec 5.1) compares the LB to baselines for using the posterior predictive to assess out-of-distribution. The Laplace Bridge tends to have better results than diagonal sampling and ties with KFAC sampling w.r.t both metrics, while being around 400 times faster than both sampling-based methods.

The second results section (Sec. 5.2) does a detailed timing comparison, indicating that computing a Hessian matrix (needed to determine the covariance \Sigma) takes less than a minute for one backward pass using the BackPACK modification of the backprop algorithm, while the training is more like 1 hour.

The final results section (Sec. 5.3) looks at a new "uncertainty-aware" top-k prediction technique, that makes use of the fact that under a Dirichlet over many classes, we can always get the marginal of a single class in closed-form as a Beta.

Strengths
---------
* Simplicity of the method
* Speed of the method (both big-oh runtime complexity and wallclock runtime wins)
* I liked Figure 3 showing the tradeoffs in KL approximation quality as number of MC samples increases
* I like the idea of uncertainty-aware top-k in Sec. 5.3. Dynamically selecting the number of "top classes" based on model predictions is a nice idea, as is the use of Beta marginals.

Weaknesses
----------
* Some typos so distracting to mislead non-expert reviewers about the fundamental method (see Eq 8)
* Other presentation issues are present throughout (e.g. often refering to primary literature instead of concisely explaining the justification for an equation)
* Should discuss connections to the logistic normal distribution
* The uncertainty-aware top-k idea needed more description in the main paper. Hiding the core method in the supplement seems to undercut its contribution.
* Proposition 1 seems distracting and not strictly needed to tell the main story of the paper


Major Presentation Concerns
---------------------------

Equations 5-7
* Need more explanation of where Eq. 7 comes from. Is there a justification for this "pseudo-inverse"?

Equation 8
* This is an approximation of another integral. I think readers would benefit from seeing this other integral written out.
* The mean parameter here is a typo, right? x is a vector of size N, but \mu_\theta has size P, so the inner product doesnt make sense as written. Shouldn't the mean parameter be f(x; \mu_\theta) ... the output of the network when weights are set to mu?
* What is the justification for Eq 8? I guess in MacKay's original derivation, the assumption here is a local linearization of the output of the network as a function of parameters (e.g. first-order Taylor approximation), and this makes the predictive distribution a Gaussian. Though do you need to model z | w, x as a random variable here to make this work? I suppose you can say this is Gaussian with vanishing small variance. Would like to see discussion improved to explain this.

Table 2 needs some clarity improvements:
* Are these timings for performing predictions for a single image? A minibatch? Please give some context.

Figure 5:
* Why are there many top 10 predictions, but not as many top 4 or top 8? I would expect a priori that these strictly decrease.


Technical Concerns
------------------

## T1: The paper seems to be unaware of the logistic normal distribution

E.g. in Sec. 5.2, the authors state "the softmax applied to a Gaussian does not have an analytic form", but should this just be the well-known logistic Normal distribution, which has a known pdf that is possible to evaluate analytically:

https://en.wikipedia.org/wiki/Logit-normal_distribution#Multivariate_generalization

Perhaps the issue is that the Logit-normal produces a K-dimensional probability vector by using K-1 dimensional means and covariances (to maintain an invertible mapping), while the present paper seems to use means/covariances of size K. Or perhaps the issue is that this logistic normal distribution does not have closed-form expressions for its mean or other moments. Either way, the discussion should be more clear.

I recommend the authors explain more carefully and cite a primary reference (e.g. J. Aitchison and S. M. Shen Biometrika 1980).


## T2: What is the point of proposition 1?

I feel that proposition 1 is added just to make the paper feel more original that just smartly using ideas from previous work.
I don't see any of the later experiments relying on this proposition. Why include it?


Experimental Concerns
---------------------

## E1: In Sec. 5.1, should we expect LB to deliver improved classification over MC with many samples?

To me, the convincing argument here is that LB is much faster than the MC sampling. I wish the experiments focused on this.

However, it seems that Table 1 is trying to make an additional point, that the LB approximation of the posterior predictive will yield *better* OOD classifier performance than MC sampling. To this I am skeptical. Why should we care that you get 0.001 better AUROC? Aren't you showing the same distribution, sampled in two different ways (each with its own possible error)?  To me, the right question to ask is, which is a better approximation of the given distribution, not which gets better OOD performance. I think the convincing thing would be to show the existing results together with a more accurate MC estimate from many more samples (100k?), and try to argue whether the LB results are a more accurate *approximation*.


Minor Line-by-line concerns
---------------------------

Between Equations 2-3:
Instead of "has to be multiplied by the Jacobian determinant", should be:
"has to be multiplied by the absolute value of the determinant of the Jacobian"

Figure 1
* Can you indicate the parameters used to generate each subfigure? Perhaps in the supplement if needed.
* Could make the point masses at the corners of the simplex a bit easier to see in the middle plots.

Figure 2 (left)
* Is the Hessian of blue and black curves p.s.d.?
* Can make clear that the black and blue dashed curves do integrate to one, just not on the interval 0-1 but over the whole real line

Sec. 5 first paragraph:
You say "First," and "Thirdly," but are missing a "Secondly," I found it hard to parse

Figure 4
* Font size on axes could be increased

---

> ### Author Response · Authors · 2020-11-11
> **Adressing comments of R1**
>
> Thank you for what we thought was an excellent review. We think that it will improve the paper both in clarity and structure. We will first address minor and then major concerns in chronological order.
>
> ### *Minor concerns:*
>
> - We have adapted the line-by-line concerns in text and figures. They can be found in the updated version of the pdf. The parameters for Figures 1 and 2 can be found in appendix A.
> - The Hessian of the blue and black curve is p.s.d. (at least at the mode). Otherwise, the Laplace approximation would be undefined (as it is in the red curve).
>
> ### *Other concerns:*
>
> - **Equation 5-7:** the entire motivation and derivation for the inverse can be found in the supplements. We have clarified this link.
> - **Equation 8:** There is indeed a typo in equation 8 and it should be $\mathcal{N}(f(x, \theta)| f_{\mu_\theta}(x), J^T \Sigma J)$. As you point out, this is a Laplace approximation of a linearized network. We have additionally added a formal statement about the Gaussian from the last-layer Laplace approximation that we use in the more complex experiments which is given by $\mathcal{N}(\mu \phi(x), \phi(x)^T \Sigma \phi(x))$ where $\phi(x)$ denotes a forward pass of $x$ through all $N-1$ layers previous to the last. Does this clarify your concerns with equation 8?
> - **T1 - Logit-Normal Distribution:** We have thought about the logit-Normal during the creation of this paper but there are two major problems with it that make it unusable. Firstly, as you already pointed out, it maps from K to K-1 but we need from K to K. Secondly, and even more crucially, it has no closed-form estimates of its mean or variance. This removes the major speed-up of the LB and thereby goes against the goal of this paper, to build a lightweight approximation. We have added the respective citation and discussion to the paper.
> - **Uncertainty-aware top-k:** We will shift the pseudocode and some of its descriptions from the supplements to the main paper if there is space left after all other clarifications. The reason why there are so many top-10 bins is that we have summarized every estimate with more than or equal to 10 classes as a top-10 estimate to not have 1000 bins in the histogram. This was described in the supplements but has now been clarified in the main text.
> - **T2 - Proposition 1:** This theoretical result is used to give a condition by which the variance of the Laplace Bridge is mapped proportionally to the corresponding Covariance matrix of the Gaussian. Intuitively speaking, it is a theoretical statement about when the LB provides a good mapping. In the supplements of the paper you can find empirical estimates of when this condition is fulfilled but independent of the empirical findings it is an important theoretical result because it shows under which circumstances the LB maps uncertainty correctly and we would therefore prefer to leave it in the paper.
> - **Table 2:** The timings are the sum of all individual minibatch (of size 128) timings for the entire CIFAR10 test set with a GPU accelerated setup. We have further clarified this in the updated version. The LB really only needs 0.014 seconds for the entire test set. This, of course, measures only the transformation from Gaussian to Dirichlet excluding the forward pass.
> - **E1 - Results of table 1:** Our goal is to show that the Laplace Bridge is significantly faster than sampling-based methods while having an approximation quality that is at least similar. So addressing your two points: a) We tried to emphasize the speed aspect as much as possible. In experiment 1, we explicitly show the computational costs of the LB vs sampling in the two right columns and experiment 2 is solely focused on timings. We have updated the text to emphasize this fact even further. b) The setup you describe, i.e. comparing a lower number of samples with a very large one (100k) is exactly what we show in the first part of experiment two (see Figure 3). If you think that this is insufficient or we have misunderstood your criticism, don’t hesitate to correct us.
>
> Thank you for your cooperative style. If you think we have misunderstood parts of your review we would be grateful for clarification. We hope our response has addressed all your concerns about our work.

---

### Decision · Program_Chairs · 2021-01-07
**Final Decision**

**Decision:**

Reject

**Comment:**

This paper proposes a mechanism for fast sampling from the posterior over the weights of the last layer of neural network, by approximating the logits as a Gaussian through equation 8. This is based on earlier work by MacKay, but has some new empirical investigations. This is a very difficult case.

In its favor:
* Despite the main ideas coming from older work by MacKay, they are interesting and relevant and worth re-surfacing.
* The experiments demonstrate some improvements in OOD detection over a diagonal Laplace approximation to the last layer, and is competitive in performance with a KFAC Laplace last layer approximation but much faster at test-time.
* The authors provided early and thoughtful responses and actively tried to have a discussion with reviewers. It is a pity that the reviewers did not participate in this discussion.

Concerns:
* While interesting, it is unclear if the proposed method actually has much practical utility in its current form. The method is presented as a fast approach for uncertainty in Bayesian deep networks. But Eq. 8 requires such significant computations to form that whatever is gained by the fast sampling may not make-up for the cost of forming the approximation itself. This computational burden is why the approach is only applied to a last layer. Table 2 and some of the surrounding discussion helps with alleviating these concerns and is most appreciated. But many basic questions persist: (i) do we really need many samples to achieve good performance, especially from a posterior over only a last layer? (we see the KL divergence decreases, but what about performance on interesting problem as a function of sample size?) (ii) in terms of total runtime-accuracy would this be competitive with using Bayesian methods over all the parameters, even if these methods are taking fewer samples? It would be easy to try. (iii) Besides OOD detection, how does this approach generally affect accuracy or calibration? (iv) How would this method compare to a basic baseline like retraining the last layer several times and ensembling? (v) could anything be done to significantly accelerate the computations in forming Eq. 8? While not all of the answers to these questions need to be favorable to the Laplace bridge for acceptance, it would certainly improve the paper to at least address most of the questions explicitly. At the end of the paper, an online setting is mentioned, which I think would be amenable to this approach --- it could be good to explore this direction.
* It is disappointing that the reviewers did not communicate with the authors, despite commendable efforts from the authors. However, the paper continued to lack a clear champion. Given the persisting lukewarm reception of reviewers,  and some of the practical concerns above, it would help to have some "stand-out" result, especially since the methodology, while interesting and relevant, is not new. That does raise some expectations for the experiments.

This is not an easy case. The paper has merits. And it's possible some of the concerns could be addressed by simply more clearly rationalizing design decisions (why would we use this approach in its current form over full Bayesian methods, which are now quite fast, with fewer posterior samples?).  At the same time it's clear the paper in its current form is not resonating with reviewers, and there are concerns about the practical applicability and limitations. It's on the borderline. Having some stand-out results could really help this paper realize its potential.